# The tetraspanin CD9 controls migration and proliferation of parietal epithelial cells and glomerular disease progression

Hélène Lazareth[1,2,3,4,18], Carole Henique [1,2,5,18], Olivia Lenoir [1,2,18], Victor G. Puelles [6,7,8,18], Martin Flamant[9], Guillaume Bollée[1,2], Cécile Fligny[1,2], Marine Camus[1,2], Lea Guyonnet[10], Corinne Millien[1,2], François Gaillard[1,2], Anna Chipont[1,2], Blaise Robin [1,2], Sylvie Fabrega[11], Neeraj Dhaun [12], Eric Camerer [1,2], Oliver Kretz[7,13], Florian Grahammer[7,13], Fabian Braun [7,13], Tobias B. Huber[7,13], Dominique Nochy[14], Chantal Mandet[14], Patrick Bruneval[14], Laurent Mesnard[15], Eric Thervet[1,2,3], Alexandre Karras[1,2,3], François Le Naour[16,19], Eric Rubinstein [17,19], Claude Boucheix [17,19], Antigoni Alexandrou[4,19], Marcus J. Moeller[6,19], Cédric Bouzigues[4] & Pierre-Louis Tharaux [1,2,3]

The mechanisms driving the development of extracapillary lesions in focal segmental glomerulosclerosis (FSGS) and crescentic glomerulonephritis (CGN) remain poorly understood. A key question is how parietal epithelial cells (PECs) invade glomerular capillaries, thereby promoting injury and kidney failure. Here we show that expression of the tetraspanin CD9 increases markedly in PECs in mouse models of CGN and FSGS, and in kidneys from individuals diagnosed with these diseases. Cd9 gene targeting in PECs prevents glomerular damage in CGN and FSGS mouse models. Mechanistically, CD9 deficiency prevents the oriented migration of PECs into the glomerular tuft and their acquisition of CD44 and β1 integrin expression. These findings highlight a critical role for de novo expression of CD9 as a common pathogenic switch driving the PEC phenotype in CGN and FSGS, while offering a potential therapeutic avenue to treat these conditions.

---

[1] Institut National de la Santé et de la Recherche Médicale (Inserm), Unit 970, Paris Cardiovascular Center – PARCC, 56 rue Leblanc, F-75015 Paris, France. [2] Université de Paris, UMR-S970, 56 rue Leblanc, F-75015 Paris, France. [3] Renal Division, Georges Pompidou European Hospital, Assistance Publique-Hôpitaux de Paris, Université de Paris, Paris F-75015, France. [4] Laboratoire d'Optique et Biosciences, Ecole polytechnique, CNRS UMR7645, INSERM U1182, Université Paris-Saclay, Palaiseau F-91128, France. [5] Institut Mondor de Recherche Biomédicale, Inserm U955, Equipe 21, Université Paris Est Créteil, Créteil F-94010, France. [6] Department of Nephrology and Clinical Immunology, University Hospital RWTH Aachen, Pauwelsstrasse 30, D-52074 Aachen, Germany. [7] Department of Medicine III, Faculty of Medicine, University Medical Center Hamburg-Eppendorf, Hamburg D-20246, Germany. [8] Department of Nephrology and Center for Inflammatory Diseases, Monash University, Melbourne VIC 3168, Australia. [9] Xavier Bichat University Hospital, Université de Paris, Paris F-75018, France. [10] National Cytometry Platform, Department of Infection and Immunity, Luxembourg Institute of Health, Luxembourg L-4354, Luxembourg. [11] Université de Paris, Institut Imagine, Plateforme Vecteurs Viraux et Transfert de Gènes, IFR94, Hôpital Necker Enfants-Malades, Paris F-75015, France. [12] Department of Renal Medicine, Royal Infirmary of Edinburgh, Edinburgh EH16 4SA Scotland, UK. [13] Renal Division, Faculty of Medicine, Medical Centre, University of Freiburg, Freiburg D-79106, Germany. [14] Department of Pathology, Georges Pompidou European Hospital, Assistance Publique-Hôpitaux de Paris, Paris F-75015, France. [15] Critical Care Nephrology and Kidney Transplantation, Hôpital Tenon, Assistance Publique-Hôpitaux de Paris, Unité Mixte de Recherche S1155, Pierre and Marie Curie University, Paris F-75020, France. [16] Inserm U1193, Université Paris-Sud, Villejuif F-94800, France. [17] Inserm U935, Université Paris-Sud, Villejuif F-94800, France. [18]These authors contributed equally: Hélène Lazareth, Carole Henique, Olivia Lenoir, Victor G. Puelles. [19]These authors jointly supervised this work: François Le Naour, Eric Rubinstein, Claude Boucheix, Antigoni Alexandrou, Marcus J. Moeller. Correspondence and requests for materials should be addressed to C.H. (email: carole.henique@inserm.fr) or to P.-L.T. (email: pierre-louis.tharaux@inserm.fr)

Necrotizing crescentic glomerulonephritis (CGN) and focal segmental glomerulosclerosis (FSGS) are life-threatening diseases leading to irreversible renal failure. Over the past years, increasing evidence supports the notion that glomerular parietal epithelial cells (PEC) are implicated in the formation of crescents in CGN and in glomerular scarring during FSGS[1–3]. In these two diseases, the common feature is the invasion of the glomerulus, the blood-filtering unit of the kidney, by PECs leading to its destruction. However, the mechanisms underlying the change in PEC phenotype remain unclear, as do those driving their oriented migration toward the glomerular tuft and their increased capacity to proliferate and form hypercellular and sclerotic lesions.

Aiming to identify factors underlying PEC activation, we revealed increased expression of CD9 using comparative deep RNA sequencing of mouse and human normal and diseased glomeruli, confirmed expression by immunohistochemistry, and investigated roles of this tetraspanin in CGN and FSGS pathogeny. CD9 has been extensively studied in cancer, where it facilitates proliferation, migration, adhesion, and survival through the organization of plasma membrane microdomains[4,5]. We hypothesized that CD9 induction in PECs may facilitate paracrine signaling across the urinary chamber, and thus constitute a pathogenic switch for glomerular demolition.

Cellular functions of CD9 are directly dependent on its cellular expression and vary according to its association with other proteins. Due to its specific structure, CD9 has the ability to interact with several partners. CD9 has been described to associate with CD44 and the pre-β1 integrin subunit[6,7], the growth factor heparin-binding EGF-like growth factor (HB-EGF)[8], as well as with epidermal growth factor receptor (EGFR)[9] and platelet-derived growth factor receptor (PDGFR)[10]. HB-EGF-dependent EGFR activation and PDGF triggering have been previously implicated in CGN and membranoproliferative glomerulonephritis[11–15], although it is unknown if they modulate the PEC phenotype.

Here, we describe the de novo expression of the tetraspanin CD9 in PEC during CGN and FSGS in mice and humans. We decipher its pathogenic role in vivo using mouse models of CGN and FSGS and show that the specific deletion of Cd9 in PEC protects from glomerular damage. In vitro, we show that CD9 deficiency impairs the ability of PECs to proliferate and to migrate toward a steep gradient of chemoattractant in microfluidic channels. We further show that high CD9 expression is required for full activation of the HB-EGF-EGFR and PDGF-BB-PDGFR pathways and thus, CD9 lowers threshold for oriented PEC migration. These findings reveal that expression of CD9 is a common pathogenic switch that directly drives glomerular injury in both CGN and FSGS in humans and mice.

## Results

**CD9 de novo expression during CGN.** In order to identify novel markers of cells participating to destructive processes in extracapillary glomerular diseases, we carried out comparative analyses of RNA sequencing data from freshly isolated glomeruli from mice with a time course after nephrotoxic serum (NTS)-induced CGN at day 4 and day 10 (Fig. 1a). This approach revealed a striking increase in Cd9 transcript abundance in glomeruli as proliferative extracapillary diseases progressed (Fig. 1b). Of note, the relative increase in Cd9 mRNA expression occurred earlier than the one of Cd44, a known marker of PEC activation[2,16–19] with an earlier significant rise, suggesting a role for CD9 in disease initiation.

Immunostaining revealed early de novo expression of CD9 in PECs and in cells invading the glomerular capillary and devoid of podocyte markers (Fig. 1c, d). This was further confirmed with observation that the majority of the CD9-expressing cells displayed CD44 and PDPN/podoplanin, whereas SNP/synaptopodin + cells expressed little or no CD9 (Fig. 1e; Supplementary Figs 1, 2). Comparative immunohistochemical analysis indicated that the prevalence of CD9 + cells consistently reflects different degrees of podocyte injury (Fig. 1d).

Immunohistochemistry analyses also supported the relevance of this finding to human crescentic glomerulonephritis. CD9 staining did not reveal glomerular expression in kidney biopsies from healthy controls or patients with proteinuric non-proliferative glomerulonephritis, such as minimal change disease (MCD). Previously reported[20–24] constitutive expression was observed in distal convoluted tubuli, collecting tubuli, and faintly in mesangial cells and vascular smooth muscle cells (Fig. 1f). In contrast, intense glomerular CD9 expression was observed in kidney biopsies from patients diagnosed with human proliferative glomerulonephritis, such as ANCA-associated nephropathy or lupus nephritis with CGN, where it was located in PECs as well as in cells engaged in destructive crescent formation (Fig. 1f).

**Cd9−/− mice are protected from CGN and extracapillary injury.** Given the strong association between de novo glomerular CD9 expression and CGN, we next investigated the role of CD9 during glomerulonephritis in the murine NTS-induced CGN model. CD9-deficient (Cd9−/−)[25] mice showed no renal phenotype at baseline. Whereas NTS-challenged Cd9+/+ mice displayed progressive deterioration of renal function, associating increased urine albumin-to-creatinine ratio (ACR) with glomerular crescentic lesions, CD9 deficiency protected mice from glomerular injury, as shown by a significantly ten-times lower ACR (440.6 ± 228.8 mg.mmol$^{-1}$ in Cd9−/− mice vs. 5204 ± 915.6 mg.mmol$^{-1}$ in Cd9+/+ mice after 21 days) and fivefold reduction of glomerular lesions (4.6 ± 3% vs. 23.9 ± 6.2% of crescents in Cd9−/− vs. Cd9+/+ mice) (Fig. 2). Strikingly, no Cd9−/− but 7/10 Cd9+/+ mice died from anuric end-stage kidney failure with 30–100% of crescentic glomeruli. CD9 deficiency conferred early protection with decreased ACR from day 5 onwards.

**Platelet and hematopoietic cell CD9 is dispensable for CGN.** The CD9 antigen is strongly expressed on platelets[26] and platelets are activated in CGN, raising the hypothesis of a role for platelet-borne CD9 in vasculitis-inducing CGN. Therefore, we generated mice with a specific deletion of Cd9 in platelets (PF4-Cd9$^{lox/lox}$ mice) by crossing Cd9-floxed mice (Supplementary Fig. 3) with mice expressing CRE selectively in platelets[27]. PF4-Cd9$^{lox/lox}$ mice were indiscernible from their control littermates (PF4-Cd9$^{wt/wt}$ mice) in terms of growth, platelet count, and renal function until the age of 6 months (Supplementary Fig. 4a–c). When challenged with the NTN model, platelet-specific CD9-deficient mice displayed similar renal injury to control mice, as measured by ACR, BUN, and crescentic glomerular lesions (Supplementary Fig. 4b–e).

CD9 was also described originally as a 24-kDa surface protein of non-T acute lymphoblastic leukemia cells and developing B lymphocytes[28], and in naive T cells[29]. We thus sought to evaluate the potential role of non-glomerular CD9 in CGN. Lethally irradiated Cd9+/+ and Cd9−/− mice were reconstituted with the bone marrow (BM) from either Cd9+/+ or Cd9−/− congenic mice. Experimental CGN was induced in the chimeric mice. Interestingly, only mice with Cd9−/− kidneys were significantly protected from severe glomerular dysfunction and crescentic demolition, whereas mice with CD9-competent kidneys were not protected from the development of glomerular lesions with no influence of the genotype of their BM cells. Notably, crescentic

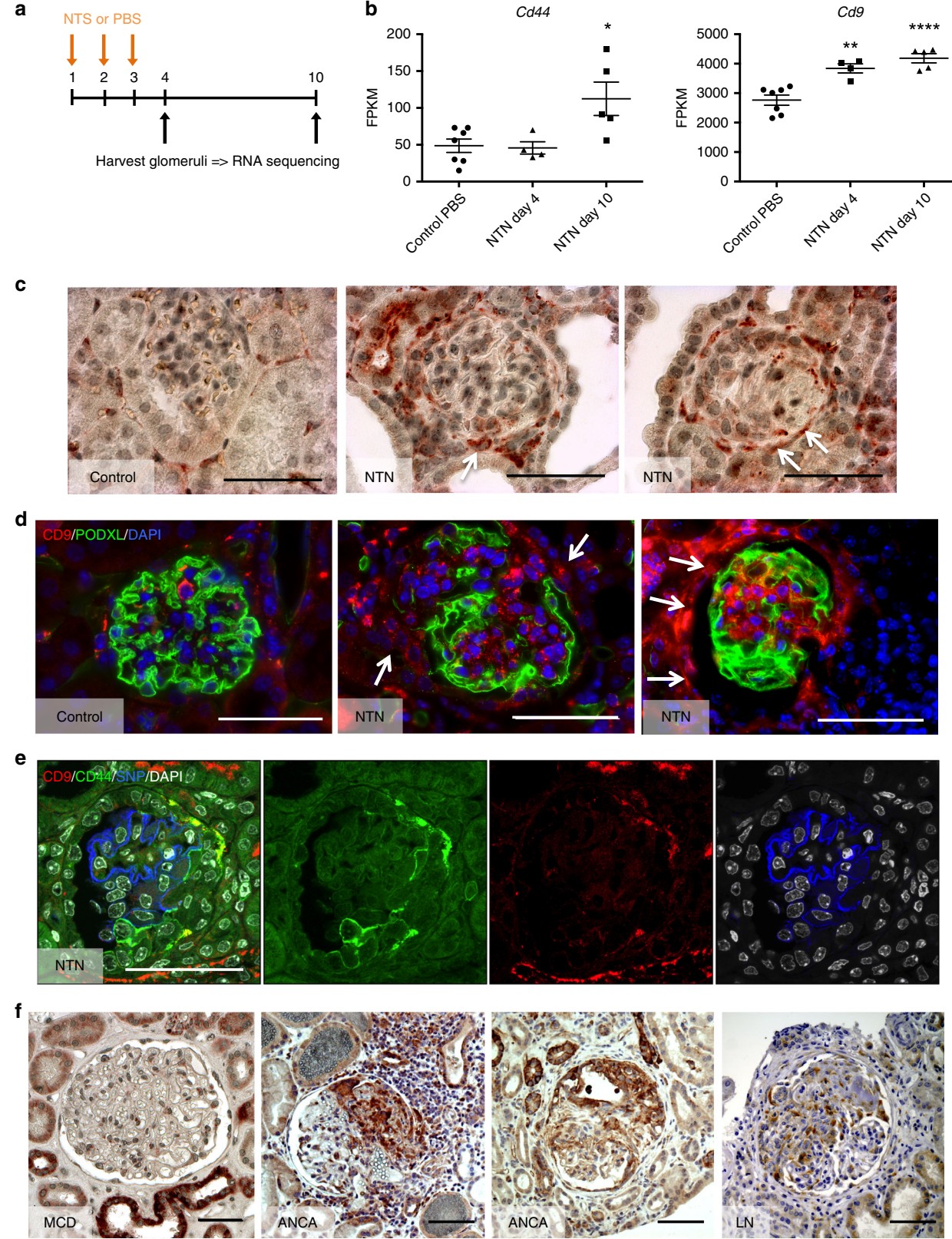

glomerular lesions were more abundant in $Cd9^{+/+}$ mice than chimeric $Cd9^{-/-}$ mice with a CD9-competent BM ($23.6 \pm 3.3\%$ vs. $8.5 \pm 1.9\%$ of crescents) (Supplementary Fig. 5).

These results support the notion that it is the glomerular expression of CD9 that participates in CGN development and that crescentic glomerular lesions are not a consequence of CD9 expression in the hematopoietic compartment.

**De novo expression of CD9 in PEC promotes glomerular lesions.** As CD9 de novo expression in glomeruli was observed in

**Fig. 1** CD9 is overexpressed by glomerular cells during CGN in mice and humans. **a** Schematic strategy to identify differentially expressed genes overtime on day 4 and 10 (black arrows), in freshly isolated glomeruli from mice developing CGN upon nephrotoxic serum (NTS) or PBS injection (orange arrows). At day 4, no crescent has been constituted yet, whereas mice display > 50% of crescentic glomeruli on day 10. **b** Relative increase in *Cd44* mRNA expression is shown as a control for assessment of PEC "activation" that was detected on day 10 only. *Cd9* mRNA expression was significantly higher in glomeruli during NTN on day 4 and 10 as compared with PBS-infused control mice. The data are expressed as mean of FPKM (fragments per kilobase per million reads mapped) $+/-$ s.e.m of four and seven mice per condition. *t* test: *$P < 0.05$; **$P < 0.01$; ****$P < 0.0001$ for NTS-injected vs. PBS-injected mice. **c** Representative images showing immunohistochemical staining of CD9 (brown) from murine kidney sections in normal conditions (control) and 10 days after nephrotoxic serum injection (NTN). Scale bar, 50 µm. **d** Representative images showing immunofluorescent stainings for CD9 (red) and PODXL/ podocalyxin (green) in adult mice at baseline (control) and 10 days after nephrotoxic serum injection (NTN). Nuclei were stained with DAPI (blue). Scale bar, 50 µm. **e** Representative images showing immunofluorescent stainings for CD9 (red), CD44 (green), SNP (blue), and DAPI (white) in adult mice at 10 days after nephrotoxic serum injection (NTN). Scale bar, 50 µm. (**f** Representative images showing immunohistochemical staining of CD9 (brown) from pathological human kidneys: MCD minimal change disease, ANCA ANCA vasculitis, LN lupus nephritis. Scale bar: 50 µm. Source data are provided as a Source Data file

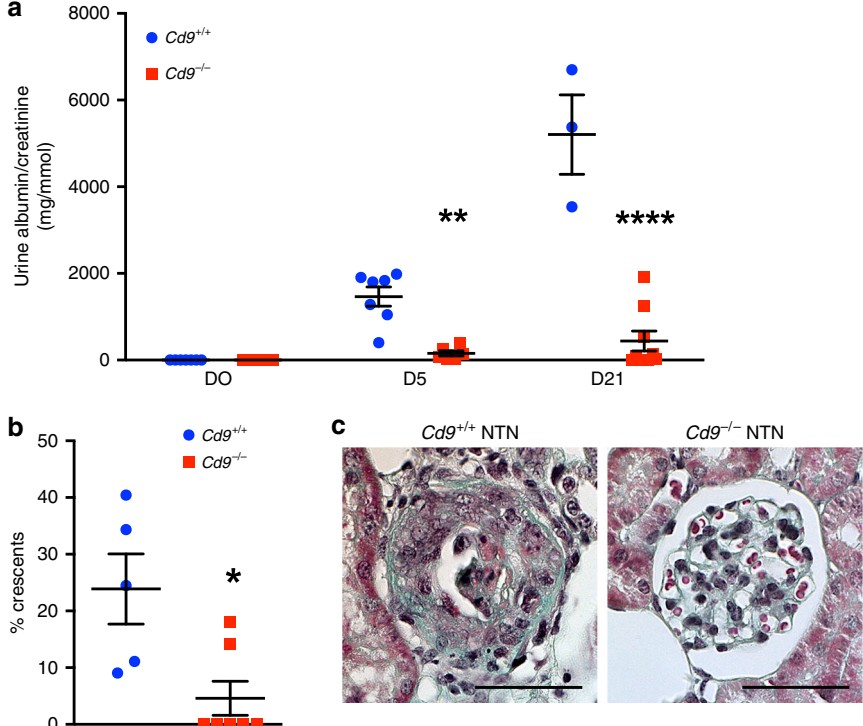

**Fig. 2** global genetic CD9 depletion in mice protects from nephrotoxic serum-induced CGN. **a** Urine albumin-to-creatinine ratio in *Cd9*$^{-/-}$ and *Cd9*$^{+/+}$ mice at baseline and during NTN model. The data represent mean $+/-$ s.e.m. of $n = 7$ and 9 mice at baseline, $n = 6$ and 9 mice at day 5 and $n = 3$ and 9 mice at day 21 (no *Cd9*$^{-/-}$ but 7/10 *Cd9*$^{+/+}$ mice had from end-stage kidney failure with 100% of crescentic glomeruli). Individual values are shown in dots. *t* test: **$P < 0.01$; ****$P < 0.0001$ for *Cd9*$^{-/-}$ vs. *Cd9*$^{+/+}$ mice. **b** Associated quantification of the percentage of glomeruli with crescent formation. The data represent mean $+/-$ s.e.m. of $n = 5$ and 7 mice per group. Individual values are shown in dots. *t* test: *$P < 0.05$. **c** Representative images showing Masson's trichrome staining on kidney sections from *Cd9*$^{-/-}$ and *Cd9*$^{+/+}$ mice 10 days after nephrotoxic serum injection (NTN). Scale bar: 50 µm. Source data are provided as a Source Data file

podocytes and PEC in human CGN and, given the recognized role of podocytes in the progression of crescentic lesions in CGN[2,30–32], we then deleted *Cd9* selectively in podocytes (Pod-*Cd9*$^{lox/lox}$ mice) by crossing *Cd9*-floxed mice with mice expressing CRE under the NPHS2 promoter (Pod-Cre mice)[33]. Pod-*Cd9*$^{lox/lox}$ mice had normal renal function at baseline, and were indiscernible from their control littermates (Pod-*Cd9*$^{wt/wt}$ mice) (Supplementary Fig. 6). In the nephrotoxic nephritis (NTN) model, Pod-*Cd9*$^{lox/lox}$ mice had similar renal injury as control mice, as shown by ACR ($807.7 \pm 184.2$ mg.mmol$^{-1}$ vs. $864.2 \pm 242.3$ mg.mmol$^{-1}$ in Pod-*Cd9*$^{wt/wt}$ vs. Pod-*Cd9*$^{lox/lox}$ mice), BUN ($51.96 \pm 9.46$ mg.dL$^{-1}$ vs. $49.37 \pm 5.01$ mg.dL$^{-1}$ in Pod-*Cd9*$^{wt/wt}$ vs. Pod-*Cd9*$^{lox/lox}$ mice), and crescentic glomerular lesions ($27.67 \pm 2.36\%$ vs. $30.00 \pm 2.63\%$ of crescents in Pod-*Cd9*$^{wt/wt}$ vs.

Pod-*Cd9*$^{lox/lox}$ mice) (Supplementary Fig. 6), thus supporting the idea that podocyte CD9 expression is not involved in crescent formation in CGN.

We then generated mice with a specific *Cd9* deletion in PEC (iPec-*Cd9*$^{lox/lox}$ mice) by crossing the *Cd9*-floxed mice with an inducible PEC-expressing CRE[34]. iPec-*Cd9*$^{lox/lox}$ mice had normal renal function (BUN = $23.9 \pm 1.3$ mg.dL$^{-1}$ vs. $20.7 \pm 1.7$ mg.dL$^{-1}$ in iPec-*Cd9*$^{wt/wt}$) and no histological abnormalities neither at a microscopic nor at an ultrastructural level. Furthermore, glomerular CD9 expression was not detected at baseline (Supplementary Fig. 7).

During the time course of the NTN model, iPec-*Cd9*$^{wt/wt}$ control mice showed a rapid increase in ACR that finally led to impaired renal function associated with severe glomerular

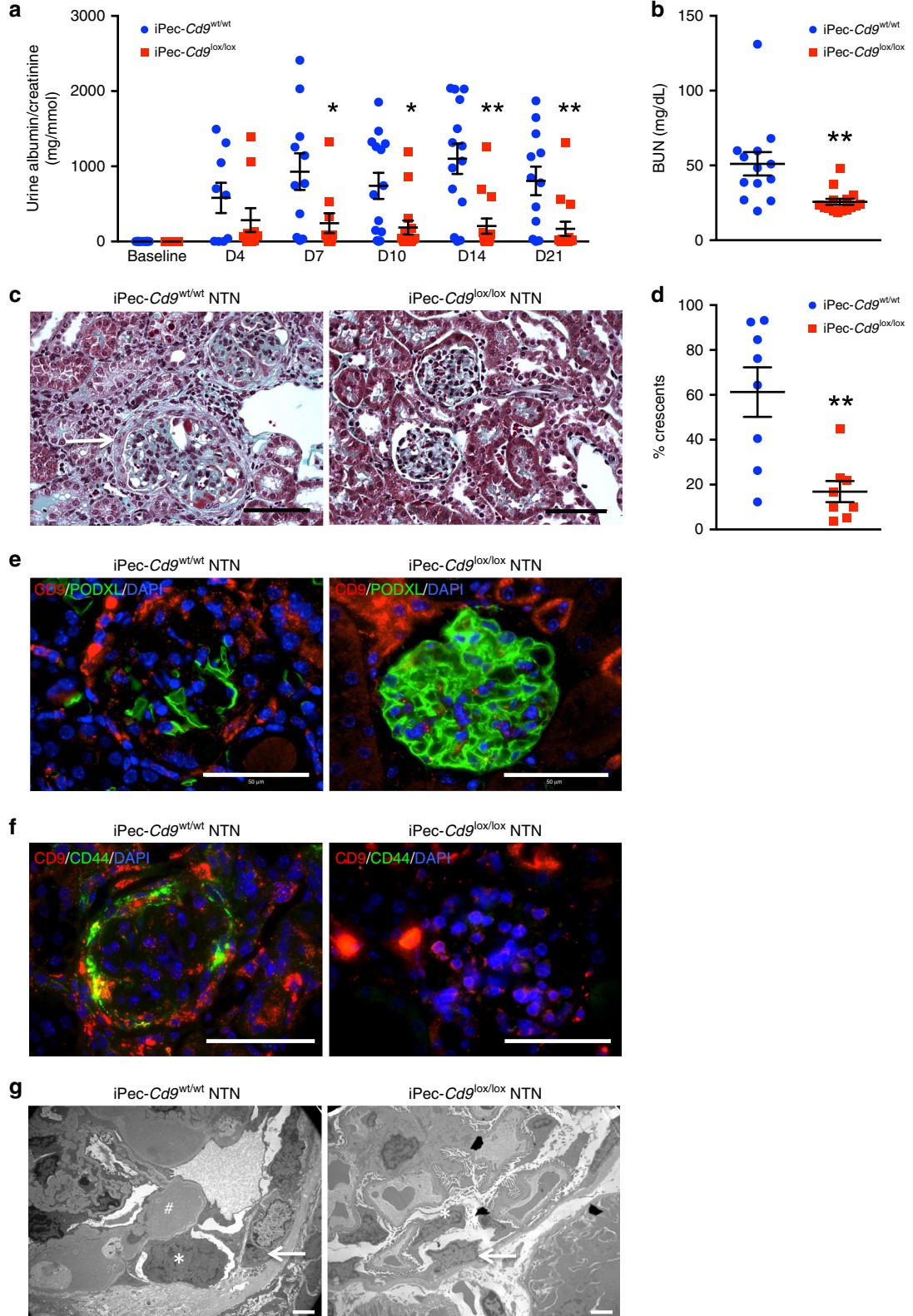

crescentic lesions (61.2 ± 11.4% vs. 16.9 ± 4.7% of crescents in iPec-*Cd9*) (Fig. 3a–d). Conversely, iPec-*Cd9*$^{lox/lox}$ mice displayed renal protection as shown by significantly lower ACR starting from day 7 of experimentation, reduced BUN and preserved renal architecture (Fig. 3a–d). Furthermore, we found CD9 expression in crescents and along the Bowman's capsule in control NTN mice, while efficient deletion of *Cd9* was validated in

iPec-*Cd9*$^{lox/lox}$ mice by no CD9 expression in PEC (Fig. 3e, f). Furthermore, while typical podocyte-PEC synechiae participating in crescent formation were observed in iPec-*Cd9*$^{wt/wt}$ NTN mice, glomerular ultrastructure was preserved in iPec-*Cd9*$^{lox/lox}$ NTN mice (Fig. 3g). Interestingly, CD9-expressing cells were activated PECs as shown by coexpression of CD9 with the hyaluronan receptor CD44[2], whereas no activated PECs were observed

**Fig. 3** PEC-selective *Cd9* deletion protects mice from nephrotoxic serum-induced CGN. **a** Urine albumin-to-creatinine ratio in iPec-*Cd9*<sup>wt/wt</sup> and iPec-*Cd9*<sup>lox/lox</sup> mice at baseline and from day (D) 4 to day 21 during NTN model. The data represent mean +/− s.e.m. of n = 14 and 15 mice per group. Individual values are shown in dots. *t* test: *P < 0.05 and **P < 0.01 iPec-*Cd9*<sup>wt/wt</sup> vs. iPec-*Cd9*<sup>lox/lox</sup> mice. **b** Blood urea nitrogen (BUN) levels in iPec-*Cd9*<sup>wt/wt</sup> and iPec-*Cd9*<sup>lox/lox</sup> mice after 21 days of the NTN model. The data represent mean +/− s.e.m. of n = 14 and 15 mice per group. Individual values are shown in dots. *t* test: ***P < 0.001. **c** Representative images showing Masson's trichrome staining of kidney sections in iPec-*Cd9*<sup>wt/wt</sup> and iPec-*Cd9*<sup>lox/lox</sup> mice after 21 days of the NTN model. iPec-*Cd9*<sup>wt/wt</sup> mice display fibrinoid necrosis (white star) and crescent (arrow). Scale bar, 50 μm. **d** Associated quantification of the percentage of glomeruli with crescents. The data represent mean +/− s.e.m. of n = 8 mice per group. Individual values are shown in dots. *t* test: **P < 0.01. **e–g** Representative images showing immunofluorescent stainings of (**e**) CD9 (red) and PODXL/podocalyxin (green) and (**f**) CD9 (red) and CD44 (green), on kidney sections from iPec *Cd9*<sup>wt/wt</sup> and iPec *Cd9*<sup>lox/lox</sup> mice after 21 days of the NTN model. Nuclei were stained with DAPI. Scale bar, 50 μm. **f** Representative images of transmission electron microscopy in iPec-*Cd9*<sup>wt/wt</sup> and iPec-*Cd9*<sup>lox/lox</sup> mice after 21 days of the NTN model. Arrows indicate parietal epithelial cells, * stand for podocytes and an # shows fibrinoid deposits. Scale bar: 2 μm. Source data are provided as a Source Data file

in iPec-*Cd9*<sup>lox/lox</sup> NTN mice (Fig. 3f). Thus, CD9 appears to be implicated in PECs activation. Altogether, these results indicate that de novo expression of CD9 in PECs participates in the development of crescents in experimental CGN.

**CD9 de novo expression in PEC during human FSGS.** Given the coexpression of CD9 with the activated-PEC marker CD44 during proliferative glomerulonephritis, we next investigated whether CD9 was involved in other diseases with PEC phenotypic changes, like FSGS[35,36]. Interestingly, CD9 was also de novo expressed in glomeruli during human FSGS, including with the collapsing variant, where strong staining was found at synechiae and along the Bowman's capsule (Fig. 4a). These observations suggest that CD9 could be involved not only in CGN progression but also in other non-proliferative glomerulopathies.

**CD9 deficiency in PEC reduces the development of FSGS.** We then challenged PEC-specific CD9-deficient mice with a model of FSGS that combines high-salt diet and deoxycorticosterone acetate (DOCA) with unilateral nephrectomy. Unine-phrectomized mice implanted with placebo pellets were used as controls. In placebo-treated wild-type mice and in iPec-*Cd9*<sup>lox/lox</sup> DOCA mice, CD9 was not detected in glomeruli, thus confirming that CD9 is not expressed in PEC at baseline (even after unine-phrectomy) and that efficient deletion in iPec-*Cd9*<sup>lox/lox</sup> mice was obtained (Fig. 4b). Conversely, in iPec-*Cd9*<sup>wt/wt</sup> DOCA mice, strong de novo glomerular expression of CD9 was observed, and co-staining with CD44 demonstrated that CD9-expressing cells were activated PEC (Fig. 4b). Vehicle-treated mice did not display renal dysfunction or histological abnormalities (Fig. 4; Supplementary Fig. 8). This supports the hypothesis that de novo CD9 expression is concomitant with PEC activation during FSGS. While iPec-*Cd9*<sup>wt/wt</sup> DOCA-salt-treated mice showed a progressive increase in ACR (164.8 ± 71.1 mg.mmol$^{-1}$ at W5 of the experiment) and BUN (25.3 ± 1.2 mg.dL) associated with glomerular sclerosis and FSGS lesions in iPec-*Cd9*<sup>wt/wt</sup> mice (percentage of glomeruli with > 50% sclerosis, 42 ± 4% and percentage of FSGS lesions, 16.9 ± 2.6% after 6 weeks of the experiment), PEC-specific CD9 deficiency was associated with a significant reduction in these parameters (ACR 86.6 ± 27.3 mg.mmol$^{-1}$ at W5 of the experiment, BUN 17.6 ± 1.1 mg.dL$^{-1}$, percentage of glomeruli with > 50% sclerosis 14.3 ± 2.3% and percentage of FSGS lesions 7.4 ± 2.6% in iPec-*Cd9*<sup>lox/lox</sup> DOCA mice at W6) (Fig. 4c–h). Not only were CD44-positive PEC observed in higher numbers in iPec-*Cd9*<sup>wt/wt</sup> DOCA mice, but the number of CD44 + PEC also strongly correlated with the number of abnormal glomeruli (R = 0.89) and glomerulosclerosis (R = 0.90) (Fig. 4i–k). CD44 + PEC surrounding the Bowman's capsule were identified in FSGS lesions by immunofluorescence (Supplementary Fig. 8a). Podocyte loss was greater in iPec-*Cd9*<sup>wt/wt</sup> DOCA mice as shown by P57 and WT1 staining and correlated with PEC

activation. Podocyte volume was increased in DOCA mice with no influence of genotype supporting the idea that the capacity of podocytes to respond to hypertrophic stress was maintained in the absence of CD9 (Supplementary Fig. 8a–g). Glomerular ultrastructure was preserved in iPec-*Cd9*<sup>lox/lox</sup> DOCA mice, whereas iPec-*Cd9*<sup>wt/wt</sup> DOCA mice exhibited large fibrous depositions in the glomerular basement membrane and flocculo-capsular synechiae (Supplementary Fig. 8h). Importantly, the differences observed between iPec-*Cd9*<sup>lox/lox</sup> and iPec-*Cd9*<sup>wt/wt</sup> mice were not related to blood pressure differences, as both genotypes demonstrated similar blood pressure increase in the time course of the DOCA-salt model (systolic blood pressure 150.6 ± 4.8 vs. 154.9 ± 4.6 mmHg in iPec-*Cd9*<sup>wt/wt</sup> vs. iPec-*Cd9*<sup>lox/lox</sup> DOCA mice) (Supplementary Fig. 9a). Hemodynamic changes cannot explain the differences in podocyte loss between iPec-*Cd9*<sup>wt/wt</sup> and iPec-*Cd9*<sup>lox/lox</sup> DOCA mice either, as glomerular volume remained unchanged between DOCA groups (8.6 ± 0.7 × 10$^5$ vs. 8.1 ± 0.7 × 10$^5$ μm$^3$ in iPec-*Cd9*<sup>wt/wt</sup> vs. iPec-*Cd9*<sup>lox/lox</sup> DOCA mice; Supplementary Fig. 9b). Interestingly, interstitial fibrosis was significantly reduced in iPec-*Cd9*<sup>lox/lox</sup> DOCA mice supporting the idea that *Cd9*-specific deletion in PEC not only influences glomerular damage but also tubulo-interstitial fibrosis and thus global kidney fate (Supplementary Fig. 9c, d). Collectively, these results support the notion that CD9 is critically involved in the development of glomerular lesions in experimental FSGS.

We thus demonstrate that CD9 correlates with PEC activation in both experimental CGN and FSGS and is required for such phenotypic switch.

We next focused on the underlying mechanisms of *Cd9*-related PEC activation using an in vitro approach.

**CD9 knockdown prevents PEC proliferation and migration.** Short-hairpin RNA interference yielded a 85% reduction in CD9 expression in a PEC cell line[37] (Supplementary Fig. 10a, b). CD9-depleted PECs displayed a strong delay in adhesion to plastic (Fig. 4a) that was associated with reduced cell spreading (3012 ± 135.7 vs. 1649 ± 70.7 μm$^2$ for scramble and *Cd9* shRNA PEC, respectively; Supplementary Fig. 10c, d). This change of phenotype was not a consequence of increased apoptosis, as measured by caspase 3-cleavage and propidium iodide/annexin-V flow cytometry (Supplementary Fig. 10e–g).

PDGF-BB stimulation induced PEC proliferation as measured by an increased number of KI67 + cells. *Cd9* knockdown reduced the number of KI67 + cells (Fig. 5b; Supplementary Fig. 10h), suggesting that CD9 controls PDGF-BB-mediated PEC proliferation. Next, we quantified in vivo PEC proliferation (PCNA + PECs) using both NTN and DOCA-salt models in iPec-*Cd9*<sup>wt/wt</sup> mice. As expected, extracapillary cell proliferation was measured in diseased iPec-*Cd9*<sup>wt/wt</sup> animals, while PEC proliferation was abolished in iPec-*Cd9*<sup>lox/lox</sup> mice (Supplementary Fig. 11).

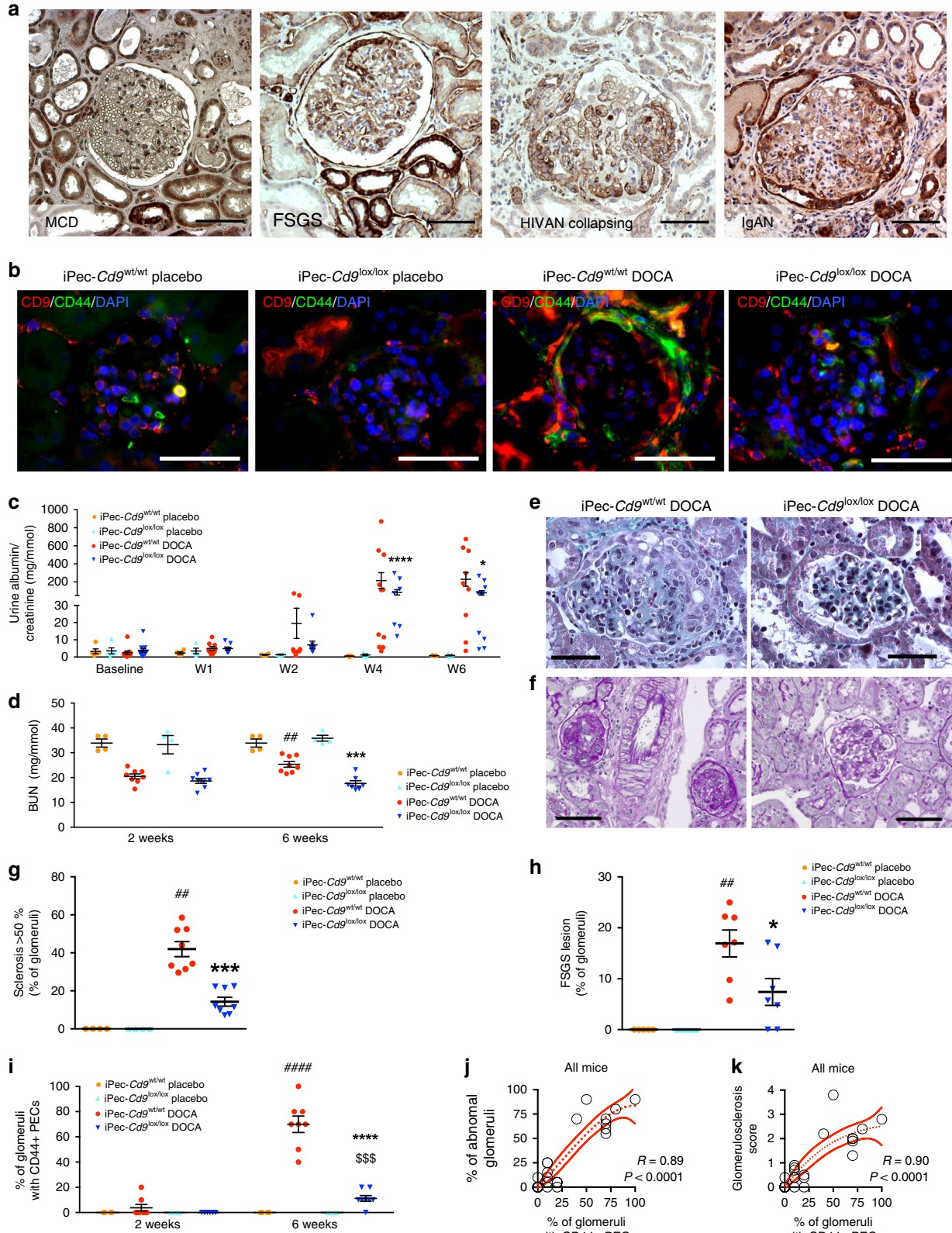

Together, these findings suggest that CD9 de novo expression may be an important mediator of PECs proliferation in FSGS and CGN.

Importantly, HB-EGF and PDGF-BB both induced PEC migration in scratch assays. *Cd9* knockdown reduced migration in response to both growth factors (Fig. 5c, d; Supplementary Movies 1–6).

**PEC sensing of local chemoattractants involves CD9.** As observed in CGN and FSGS, PECs attach to and migrate toward the glomerular tuft. This led us to hypothesize that they may be sensitive to chemoattractants filtered through the capillary and/or locally produced by it.

Therefore, we modeled a growth factor steep gradient in microfluidic channels to evaluate the capability of PECs to sense

**Fig. 4** Glomerular cells overexpress CD9 during FSGS and PEC-selective *Cd9* deletion protects mice from FSGS in the DOCA-salt model. **a** Representative images showing immunohistochemical staining of CD9 (brown) from pathological human kidneys: minimal change disease (MCD), primary FSGS, collapsing FSGS in HIV-associated nephropathy (HIVAN), and collapsing FSGS in IgA nephropathy (IgAN). Scale bar, 50 μm. **b** Representative images showing immunofluorescent stainings of CD9 (red), PODXL/podocalyxin (green), and DAPI (blue) on kidney sections from iPec-*Cd9*wt/wt and iPec-*Cd9*lox/lox mice at week 6 of the DOCA-salt model. Scale bar, 50 μm. **c** Urine albumin-to-creatinine ratio and (**d**) BUN in iPec-*Cd9* mice during the DOCA-salt model ($n = 5$ mice in placebo groups, $n = 10$ mice in DOCA groups); ANOVA: *$P < 0.05$, ***$P < 0.001$, and ****$P < 0.0001$ between iPec-*Cd9*wt/wt and iPec-*Cd9*lox/lox mice in the DOCA-salt model; ##$P < 0.01$ between W2 and W6 of DOCA treatment in iPec-*Cd9*wt/wt group. **e**, **f** Representative images showing Masson's trichrome (**e**) and periodic-acid Shiff (**f**) staining on kidney sections from iPec-*Cd9*wt/wt and iPec-*Cd9*lox/lox mice after 6 weeks of DOCA. Scale bar, 50 μm. **g**, **h** Associated quantification of the percentage of glomeruli (**g**) with more than 50% of sclerosis and (**h**) with FSGS lesions ($n = 5$ and 8 mice, respectively, in placebo and DOCA groups). *iPec-*Cd9*wt/wt vs. iPec-*Cd9*lox/lox mice in the DOCA groups; #iPec-*Cd9*wt/wt placebo-treated vs. DOCA-treated; ANOVA: *$P < 0.05$, ##$P < 0.01$, ***$P < 0.001$. **i** Quantification of the percentage of glomeruli with CD44-positive PEC at W2 and W6 in iPec-*Cd9*wt/wt and iPec-*Cd9*lox/lox mice ($n = 4$ and 8 mice, respectively, in placebo and DOCA groups). *iPec-*Cd9*wt/wt vs. iPec-*Cd9*lox/lox mice in the DOCA groups; #iPec-*Cd9*wt/wt placebo-treated vs. DOCA-treated; $iPec-*Cd9*lox/lox placebo-treated vs. DOCA-treated. ANOVA: $$$$P < 0.001$, **** and ####$P < 0.0001$. **j**, **k** Correlation between the percentage of CD44-positive PEC and the percentage of abnormal glomeruli (**j**) or glomerulosclerosis score (**k**) in all mice. Source data are provided as a Source Data file

local changes in chemoattractants that may surround them in the urinary chamber.

We used microfluidic microchannels to assess the oriented migration of PECs in a PDGF-BB gradient (Fig. 5e–g). The T-shape of the microfluidic microchannel allowed us to apply PDGF-BB-containing medium at one entrance and standard medium at the other, thus creating a PDGF-BB concentration gradient in the principal branch of the channel. The shape and the values of the gradient could be calculated at every position (*y* axis, Fig. 5g). The absolute value of the average cell displacement was similar between scramble and *Cd9* shRNA PECs (Fig. 5h), suggesting that the ability to migrate was preserved despite defects in adhesion and spreading observed under static conditions. PECs showed limited motion with the direction flow, and instead chose to migrate orthogonally to the flow attracted by the PDGF-BB gradient. No migration or enhancement of motility were observed in regions of the microchannels where the average PDGF concentration is significantly higher (Supplementary Fig. 12), suggesting that the effect of CD9 knockdown cannot be compensated by significant PDGFR overstimulation. Strikingly, CD9 depletion abolished the ability of PECs to respond to the gradient (Fig. 5i). Altogether, these experiments indicate that *Cd9* knockdown in PECs impairs their gradient-sensing ability without affecting their motility.

**CD9-deficient PECs display reduced expression of β1 integrin.** Given the role of ITGB1 in cell spreading and migration[38–40] and the fact that CD9 has been found to associate with the pre-β1 subunit of ITGB1 in other cell types[41], we then analyzed ITGB1 expression in CD9-depleted PECs. Knocking down *Cd9* in PECs resulted in an ~30% reduction in ITGB1 content in cell lysates (Fig. 6a, b), while immunofluorescence confirmed loss of ITGB1 membrane expression in CD9-deficient PECs (Supplementary Fig. 13a). Not only *Itgb1* but also *Itgb3*, *Itga1*, *Itga3*, and *Itga10* mRNA expression were decreased in CD9-deficient PECs, thus suggesting involvement in altered cytoskeletal dynamics in these cells (Supplementary Fig. 13b). In situ, basal ITGB1 expression was low with a mesangial and endothelial endocapillary pattern (Supplementary Fig. 14). ITGB1 expression was enhanced in iPEc-*Cd9*wt/wt diseased glomeruli in both experimental CGN and FSGS and localized to crescents and activated PECs. Interestingly, expansion of the ITGB1 expressing cells correlated with loss of podocyte marker NPHS2 (Supplementary Fig. 14) and colocalized with PEC activation marker CD44 to a large extent, but displayed a more widespread expression (Supplementary Fig. 15).

In iPec-*Cd9*lox/lox NTN and iPec-*Cd9*lox/lox DOCA mice, ITGB1 expression was low in PEC (Fig. 6c; Supplementary

Fig. 14). These results suggest that ITGB1 is a novel marker of PEC activation and might participate in glomerular extracapillary lesions formation.

**Podocyte loss is not sufficient to trigger CD9 expression in PECs and FSGS.** It has been believed that podocyte injury may occur prior to the PEC activation. Therefore, we evaluated whether podocyte injury would induce CD9 expression in PECs. To this end, we assessed CD9 glomerular expression in a model characterized with accentuated podocyte loss and glomerulosclerosis upon genetic targeting of podocyte autophagy that exacerbates diabetic nephropathy. In that model, we demonstrated increased proteinuria and podocyte injury with foot process effacement and loss of differentiation markers in *Nphs2*. cre *Atg5* lox/lox diabetic mice, but never observed FSGS nor synechiae [42]. We re-analyzed these kidneys looking for the PEC activation marker CD44 and for CD9 expression. We found no CD44 and no CD9 expression in PECs, neither in diabetic WT nor in diabetic *Nphs2*.cre *Atg5*lox/lox mice (Supplementary Fig. 16), despite the latter group showed marked podocyte loss. Thus, in this specific case, podocyte injury is not sufficient to induce neither PEC activation nor CD9 de novo expression.

Conversely, we recently observed experimental FSGS without primary podocyte insult but with manipulation of the endothelial HIF2/EPAS1 pathway using mouse genetics [43]. Upon chronic angiotensin II infusion and high salt diet, *Cdh5*-CRE *Epas1*lox/lox mice developed similar degree of podocyte injury to their wild-type counterparts. Surprisingly, FSGS lesions with CD44 + and fibronectin + PECs were observed only in *Cdh5*-CRE *Epas1*lox/lox hypertensive mice [43]. We next analyzed CD9 expression and found de novo CD9 expression in PECs in the hypertensive *Cdh5*-CRE *Epas1*lox/lox mice only. De novo high CD9 expression was almost exclusively associated with FSGS lesions and CD44 expression as shown in Supplementary Fig. 16. Altogether, these findings suggest that endothelial derived mediators may contribute to CD9 induction in PECs and FSGS.

**PDGFR and EGFR pathways are impaired in *Cd9*-deficient PEC.** PDGF-BB-mediated cell migration involves signal transduction through PDGFRβ phosphorylation, notably at Tyr 1009[44], and subsequent FAK activation. Interestingly, *Cd9*-depleted PECs showed decreased abundance of PDGFRβ both at protein and mRNA level (Supplementary Fig. 13c). While PDGF-BB induced a rapid and transient PDGFRβ phosphorylation, *Cd9*-depleted PECs showed impaired PDGFRβ phosphorylation (Fig. 6d–f). As a consequence, activation of FAK was also

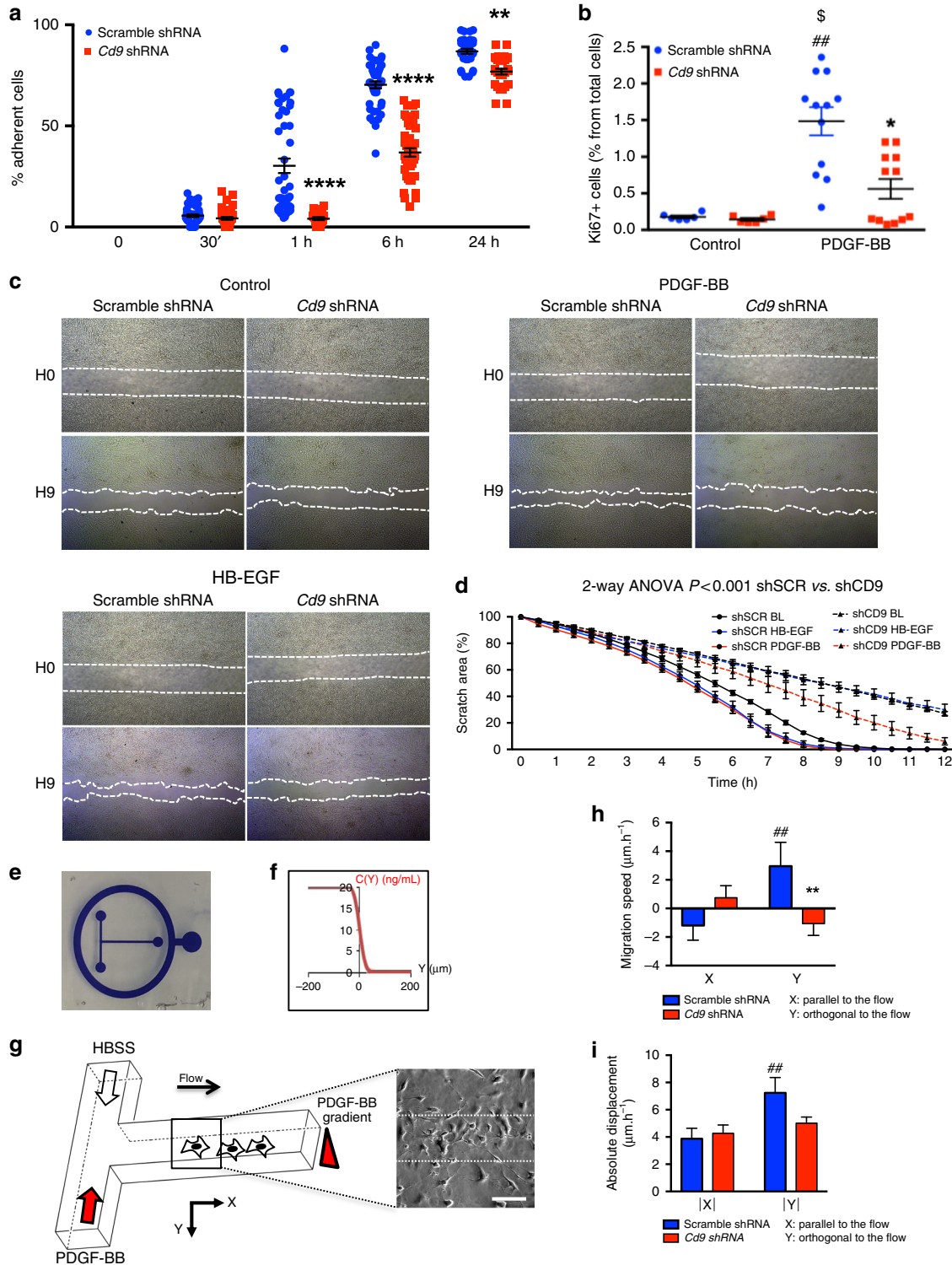

impaired, as shown by reduced phosphorylation at Tyr 397 (Fig. 6d, g). In a similar manner, we observed that the HB-EGF/ EGFR/FAK pathway was also defective in CD9-depleted cells with a decreased level of total EGFR and its Tyr 1068 phosphorylation after HB-EGF stimulation. Consistently, FAK phosphorylation was reduced as well (Fig. 6h–k; Supplementary Fig. 13c). These results suggest that the impaired migratory capacity observed in *Cd9*-depleted PECs is a consequence of defective PDGFRβ and EGFR signaling. Indeed, the lack of gradient-sensing ability (chemotaxis) of CD9-depleted PECs cannot be explained by the

observed decrease in ITGB1 levels only, as cell motility is preserved.

## CD9 expression in humans correlates with CD44 and ITGB1.

As we observed that CD9 expression in PEC correlated with PEC activation in experimental CGN and FSGS, and that CD9 depletion was associated with decreased ITGB1 levels, we analyzed CD9 expression in combination with CD44 and ITGB1 in human glomerulopathies. Patient characteristics are displayed in

**Fig. 5** In vitro CD9 depletion alters adherence, proliferation, and migration in PEC. **a** Percentage of adherent cells from 30 min (30') to 24 h in PEC transduced with a *Cd9* shRNA-coding lentivirus or a scramble shRNA-coding lentivirus. The data are shown as the mean +/− s.e.m. of $n = 25$–40 wells per condition. *t* test: **$P < 0.01$, ***$P < 0.0001$. **b** Cell proliferation of control PEC (scramble shRNA) and CD9-depleted PEC (*Cd9* shRNA) assessed by flow-cytometry measurement of the percentage of KI67 positive cells under basal conditions and after PDGF-BB stimulation. *t* test: $ scramble shRNA vs. scramble shRNA with PDGF-BB, $^{\$}P > 0.05$; #*Cd9* shRNA in basal condition vs. scramble shRNA after PDGF-BB stimulation, ##$P < 0.01$; *scramble shRNA vs. *Cd9* shRNA after PDGF-BB stimulation, *$P < 0.05$. The data are shown as the mean +/− s.e.m., and individual values are shown in dots ($n = 6$ and 12 wells per condition). **c** Migration chambers showing area of migration of scramble shRNA and *Cd9* shRNA PEC in basal conditions (control) or after stimulation by HB-EGF or PDGF-BB during 9 h. **d** Quantification of the area of migration from $t = 0$ (100%) to $t = 12$ h. Two-way ANOVA, ***$P < 0.001$, *Cd9* shRNA (shCD9) compared with scramble shRNA (shSCR) under stimulation with HB-EGF or with PDGF. The data are shown as the mean +/− s.e.m. of $n = 4$ chambers per condition. **e** Picture of a microfluidic device. **f** Shape of the gradient in the microchannel. Concentration ($C(Y)$) at different coordinates of Y are depicted. **g** Schematic representation of T-shape microfluidic channel showing two entrances perfused either with the PDGF-BB or standard HBSS medium. **h, i** Quantification of (**h**) the oriented migration and (**i**) of the absolute displacement, of control (scramble shRNA) and CD9-depleted (*Cd9* shRNA) PEC after stimulation with a gradient of PDGF-BB (20 ng/mL during 3 h). *t* test: ##$P < 0.01$ between X (displacement in the flow) and Y (displacement in the gradient) in basal PEC; **$P < 0.01$ between scramble shRNA and *Cd9* shRNA in Y. The data are shown as the mean +/− s.e.m. of $n >$ 30 cells in 3 or 4 T-shape microfluidic channel per condition. Source data are provided as a Source Data file

Supplementary Tables 1 and 2. We found that CD9 closely colocalized with ITGB1 during human extracapillary glomerulopathies such as ANCA-associated CGN and FSGS, but not in non-proliferative glomerulonephritides or in normal kidney (Fig. 7). Notably, CD9 and ITGB1 are predominantly expressed by PECs along the Bowman's capsule in FSGS lesions and in crescents. CD9-CD44 co-staining further confirmed that activated PECs account for a large number of CD9-expressing cells. These data support the hypothesis that de novo CD9 expression in CGN and FSGS contributes to the formation of glomerular lesions (i.e. crescent formation or synechiae) through PEC migration/activation, involving HB-EGF-EGFR and PDGFR pathway activation, and increased ITGB1 levels.

## Discussion

Here, we report that pathogenic expression of CD9 by glomerular parietal epithelial cells drives glomerular damage during CGN and FSGS. This tetraspanin has emerged as a threshold determinant from our search of systematic clusters of epithelial cell surface components capable of transducing migratory and proliferative signals in epithelial cells.

Although PECs have been implicated in the generation of fibrotic lesions in FSGS[45] and in the formation of the crescent in CGN[34], we provide evidences that PEC protein is directly implicated in their activation and pathogenic transformation of glomerular structure and function. Using genetic tracing of glomerular epithelial cells, Smeets et al. previously showed that PEC accounts for the largest number of cells in the formation of the crescent during CGN and FSGS in mice[2]. Here, we show that CD9 is de novo expressed in PECs among renal pathologies involving PEC activation (i.e migration and proliferation). Interestingly, CD9 expression in PECs was found in FSGS-like (DOCA-salt and angiotensin II-induced hypertension in a sensitive genetic model[43]) and CGN-like rodent models, always in association with CD44 expression. Conversely, no CD9 expression was found in PECs in mouse model of diabetic nephropathy, even in a model prone to podocyte injury (i.e. *Nphs2*.cre *Atg5*[lox/lox]). Thus, CD9 seems to be a novel and selective driver of PEC activation.

Furthermore, we demonstrate that a specific ablation of CD9 expression in PECs confers protection not only in the inflammatory model of CGN but also in a model of FSGS, a more chronic form of glomerular injury.

Podocyte injury is critical for destruction of the filtration barrier in glomerulonephritis. Silencing *Cd9* not only reduced PEC-induced glomerulosclerosis but also maintained the number of podocytes. While CD9 is expressed by both podocytes and PECs in human pathologies, CD9 de novo expression in

podocytes in NTN and DOCA-salt models was low, probably explaining why *Cd9* deletion in podocytes had no impact on outcomes.

This supports the notion that podocyte displacement by invading PECs contributes to the destruction of the glomerular filtration barrier in FSGS. Triggers for CD9 overexpression are unknown. CD9 was found to be upregulated by mechanical stress in cultured immortalized podocytes whereas most of the tetraspanins remained unaffected[46]. Thus, further work would be useful to ascertain the influence of mechanical stimuli in CD9 regulation in PECs.

Changes in the PEC phenotype, i.e. the proliferation and migration of a normally quiescent cell population, represent key steps in the destruction of glomeruli during CGN and FSGS[1,34,45]. It has been previously shown that CD9 can either promote or suppress cancer cell migration and metastasis, depending on the type of cancer, the type of cells involved, and the migratory signal[47,48]. Meanwhile the case of many types of cancer cells, where CD9 promotes cell proliferation and migration[49–51], may share similarities with the mechanisms, whereby CD9 drives renal disease via aberrant expression in PECs. Targeting CD9 with specific antibodies can reduce the migration of malignant cancer cells[52]. Concordantly, *Cd9* silencing in immortalized PECs impaired their ability to proliferate and migrate in a directional manner. These findings should help decipher the origins of glomerular injury. Interestingly, CD9 induces a complete remodeling of PECs and seems to drive an EMT-like phenotype as shown by changes in *Cldn1, Vimentin,* or *Snail* in CD9-deficient PECs.

How does CD9 expression help drive PEC out of quiescence? HB-EGF activation of EGFR is a key driver of renal damage in early stages of mouse and human glomerulonephritis[11], and CD9 is known to form molecular microdomains at the cell surface in which EGFR is associated[9] and modulate its activation by yet unknown mechanisms. After paracrine activation of the EGFR pathway by TGFα, CD9 potentiates EGFR signaling[53], although we previously ruled out the involvement of TGFα in CGN[11]. Interestingly, CD9 was shown to be a « diphtheria toxin receptor-associated protein » (DRAP27), forming a complex with proHB-EGF at the cell membrane, and upregulating its juxtacrine mitogenic activity[54]. Although this could likely play a pathogenic role, we suspect that other complex mechanisms are at play to explain the powerful effect of CD9 deficiency in vivo and in vitro in the presence of soluble chemoattractants such as mature HB-EGF and PDGFBB.

The PDGFR pathway also contributes to renal damage during CGN. Upregulation or de novo expression of PDGF-BB has been described in mesangial cells, podocyte vascular cells, tubular and interstitial cells in animal models, and human renal diseases in

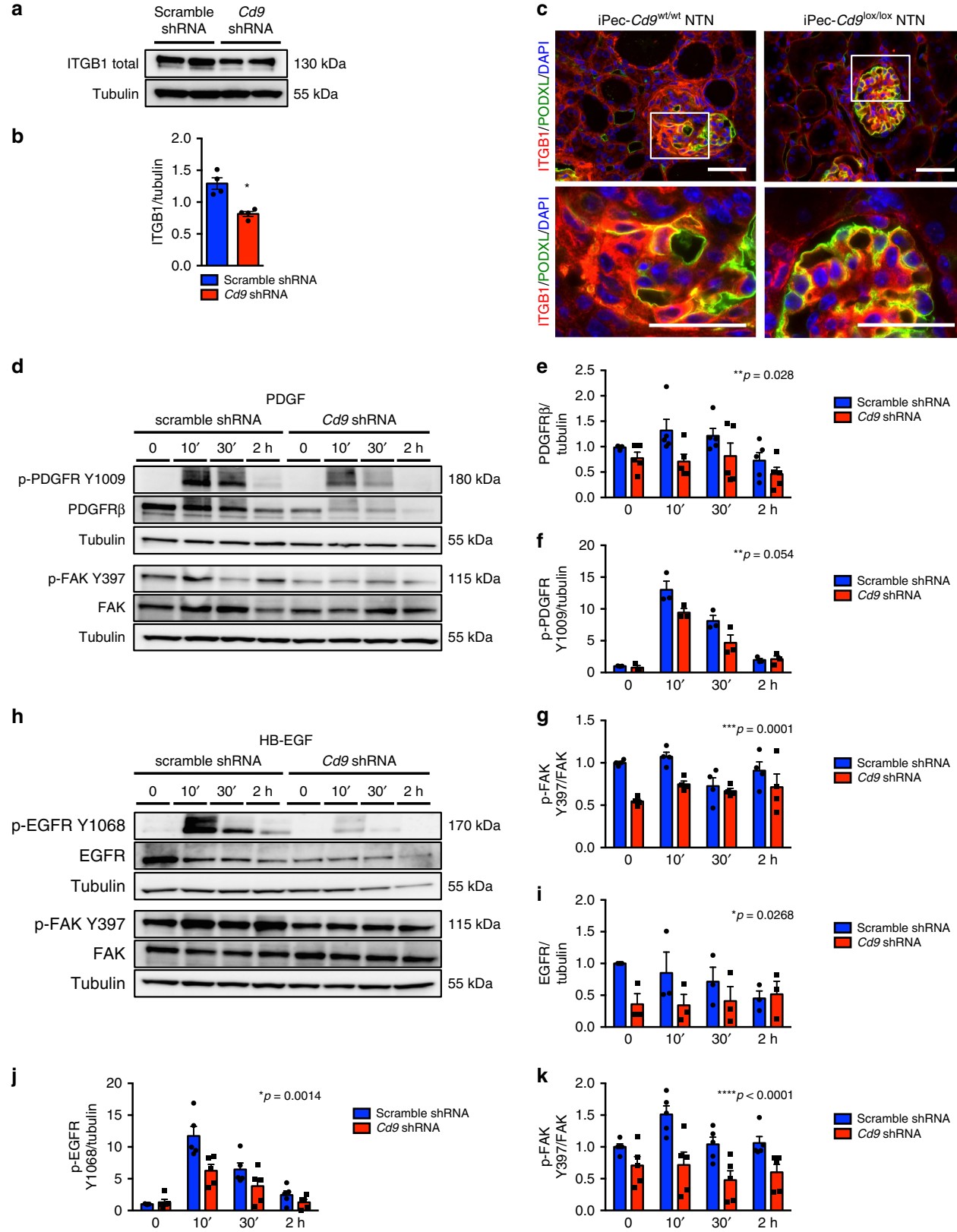

many studies[55]. The PDGFR has also been reported to associate with CD9[10], and PDGFRβ levels and activation were indeed reduced upon CD9 depletion. Potentiation of EGFR and PDGFR pathways would be predicted to enable chemotaxis by increasing engagement of downstream effectors, such as FAK. Various RTKs are recruited to and enriched within specific plasma membrane

microdomains[56], and CD9 tetraspanin-enriched microdomains may control actin-dependent protrusive membrane microdomains, such as dorsal ruffles and invadosomes. CD9 may thus promote growth factor signaling through the formation and stabilization of tetraspanin-enriched signaling microdomains[57] that promote EGFR and PDGFRβ receptor insertion in the

**Fig. 6** CD9 depletion alters EGFR- and PDGF-dependent FAK activation and reduced ITGB1 expression. **a** Western blot analysis of the expression of ITGB1/β1 integrin in control (scramble shRNA) and CD9-depleted (*Cd9* shRNA) PEC and (**b**) quantification. Tubulin was used as a loading control. **c** Representative images of immunofluorescent stainings of ITGB1 (red) and PODXL/podocalyxin (green) in glomeruli from iPec-*Cd9*^wt/wt and iPec-*Cd9*^lox/lox mice after 14 days of the NTN model. Scale bar, 50 μm. Nuclei were stained with DAPI (blue). **d** Western blot analysis of the expression of phospho-PDGFR (Y1009), PDGFR, phospho-FAK Y397, and FAK in control (scramble shRNA) and CD9-depleted (*Cd9* shRNA) PEC after time sequential stimulation with PDGF-BB and (**e–g**) quantifications. Tubulin was used as a loading control. **h** Western blot analysis of the expression of phospho-EGFR Y1068, EGFR, phospho-FAK Y397, and FAK in control (scramble shRNA) and CD9-depleted (*Cd9* shRNA) PEC after time sequential stimulation with HB-EGF and (**i–k**) quantifications. Tubulin was used as a loading control. **e–g, i–k** The data represent mean $+/-$ s.e.m. of $n = 4$ experiments. *$P < 0.05$ scramble shRNA vs. *Cd9* shRNA using two-way ANOVA test. Source data are provided as a source data file

plasma cell membrane. Its facilitating role upstream of both HBEGF-EGFR and PDGFR pathways could position CD9 as an attractive therapeutic target for both CGN and FSGS.

Another relevant mechanism of action of CD9 could be through modulation of integrin signaling. In cells that have an abundant pool of intracellular integrins, CD9 (but not CD81 or CD82) is associated with the pre-β1 subunit and calnexin, an ER chaperone protein[41]. Whether or not these early associations with tetraspanins are required for proper biogenesis or turnover of the integrin remains to be determined. This is suggested by the decrease in ITGB1 protein abundance in *Cd9*-deleted PEC that could in turn contribute to their lack of adhesion and motility. The report of a CD9 conformation-dependent epitope whose expression depends on changes in the activation state of associated α6β1 integrin suggests another potential level of functionally relevant CD9 involvement in β1 integrin-dependent cellular processes[58].

Reduced ITGB1 expression could also participate in the decrease of PEC proliferation as integrin-mediated extracellular-matrix attachment is crucial to ensure glomerular epithelial cells response to growth factors[59]. Increased expression of both β1 and β3 integrins have been reported in human CGN[60,61]. In rodent models, β1 integrin is overexpressed 7 days after CGN induction, promoting cell adhesion to the matrix proteins[62]. Recently, Prakoura et al. described the colocalization of NF-κB-induced periostin with β3 integrin in activated crescentic PEC in the NTN model[61]. As a major laminin and collagen receptor, integrin β1 promotes glomerulosclerosis during FSGS by driving collagen production[63]. Integrin-linked kinase, which interacts with β1 integrins, is overexpressed by PGP.5-positive cells (PGP.5 being a PEC marker) in the crescent during rat CGN[64].

CD44 has been found recently to participate in PECs proliferation in experimental CGN and FSGS[18]. Thus, CD9 is a major regulator of PEC activation also through control of CD44 expression as CD9-deficient PECs presented decreased *Cd44* mRNA expression (Supplementary Fig. 13c) and more spectacularly, no induction of CD44 high expression upon NTS or DOCA-salt UNx challenge (Figs 3, 4).

Key observations in rodent models could be reproduced in human glomerulopathies, suggesting a role for CD9 also in human disease. Indeed, during human CGN and FSGS, we showed that activated PEC expressed CD9 that colocalized closely with ITGB1 and CD44, especially in crescents and synechiae across the urinary chamber. Interestingly, occurrence of combined expression of CD9 with ITGB1 was more consistent than combined expression of CD9 with CD44, suggesting the existence of various stages of PEC activation or distinct subsets of such cells. Altogether, our results support the implication of CD9 in PEC activation and ITGB1-mediated glomerular lesions in both diseases. Although we demonstrate that a clear interaction exists between CD9, ITGB1, and EGFR/PDGFR, the precise way by which these proteins interact to regulate adhesion and chemotaxis is complex, possibly multifactorial and requires further investigations.

At last, we observed that PDGF-BB and HB-EGF that are produced by the glomerular tuft in CGN, and to a lesser although significant extent in FSGS, display chemotactic and mitogenic influences on PECs. Modeling urinary gradients of PDGF-BB or HB-EGF in microfluidic channels demonstrated the capability of PECs to sense local changes in chemoattractants in the urinary chamber and revealed a critical facilitating role for CD9 in this process. Extension of such proof-of-concept experiments should help identifying other key molecules emanating from the injured capillary and triggering PEC-mediated maladaptive response to injury.

In conclusion, we demonstrate that preventing local expression of CD9 by PEC in glomeruli alleviates glomerular damages in two distinct severe diseases, CGN and FSGS (Supplementary Fig. 17). Targeting the CD9 pathway could offer therapeutic options and warrants further attention. CD9 expression in PECs could also provide a marker for diagnosis of CGN and FSGS. Indeed, CD9-positive PECs involved in crescent and sclerotic lesions were not always CD44 positive, implying that markers of PEC activation may vary as a function of time or reflect heterogeneous PEC subsets. Thus, CD9 staining may be considered for early diagnosis and management of these diseases.

Finally, our study suggests a concept in which severe kidney diseases characterized by pathogenic PEC recruitment to the glomerular tuft may all depend on a CD9-dependent molecular complex that adjusts the threshold for proliferation and directional migration of these cells.

## Methods

**Animals.** Mice with a constitutive CD9 deficiency have been previously described[25]. For the generation of *Cd9*-floxed mice, we inserted on the PTV-0 vector with a floxed *Neo*^r gene (i) 3′ to the *Neo*^r gene a 9 -kb genomic fragment containing *Cd9* exon 1 (starting at bp -1625 from the ATG site) and a *lox* site introduced at the first *Nhe*I site of *Cd9* intron 1 (3.6 kb 3′ from the start of the 9- kb fragment); and (ii) 5′ to the *Neo*^r gene a short arm of 787 bp. The recombination was performed in E14 ES cells (Supplementary Fig. 3). One of the selected ES clones, RH289, was transfected with the expression vector pIC-Cre[65] to remove the *Neo*^r gene. The screening of recombinant ES clones was performed by PCR, and the recombination was checked by southern blot. E14 Clone RH289-140 harboring the expected recombination was used for injection into 3.5-day-old C57BL/6 blastocysts, which were transferred into foster mothers. Chimeric males were crossed with C57Bl/6 females, and heterozygous mice were intercrossed to check for normal CD9 expression of homozygous mice. Heterozygous floxed mice were backcrossed 15 times on a C57Bl/6J background. Primers used for genotyping the floxed *Cd9* gene and the floxed deleted gene (*Cd9*-fdel) are: UP1 5′-TGCAGGCATGGAGGCGCA GC-3′, LO1 5′-GTGCCGGCCTCGCCTTTCCC-3′, m*Cd9* 5′-CTGGTCACACCCC CTAACGGAGC-3′, NL1 5′-CCTACATCTCCCATCTGCCCCCCAT-3′, NL2 5′-C ATGGAGCTTGGGGAGGCCTTTGGA-3′. CRE recombination deletes the same genomic fragment than the constitutive *Cd9* gene deletion previously reported and completely suppressed CD9 expression[25] (Supplementary Fig. 3).

Mice with a specific deletion of *Cd9* in platelets (PF4-Cre-*Cd9)* were generated by crossing homozygous *Cd9* flox/flox mice with PF4-Cre mice[27]. Mice with a podocyte-specific deletion of *Cd9* (Pod-*Cd9*) were generated by using *Nphs2*-Cre recombinase, which expresses CRE recombinase exclusively in podocytes[33].

Mice with PEC-specific deletion of *Cd9* were generated by crossing iPec-Cre-positive mice, which expresses CRE recombinase exclusively in PEC after doxycycline induction, with *Cd9*-floxed mice[34]. iPec-Cre mice were on a mixed background (C57BL6/J and SV129). Briefly, inducible CRE expression in PEC was generated by crossing mice with the enhanced reverse tetracycline transactivator

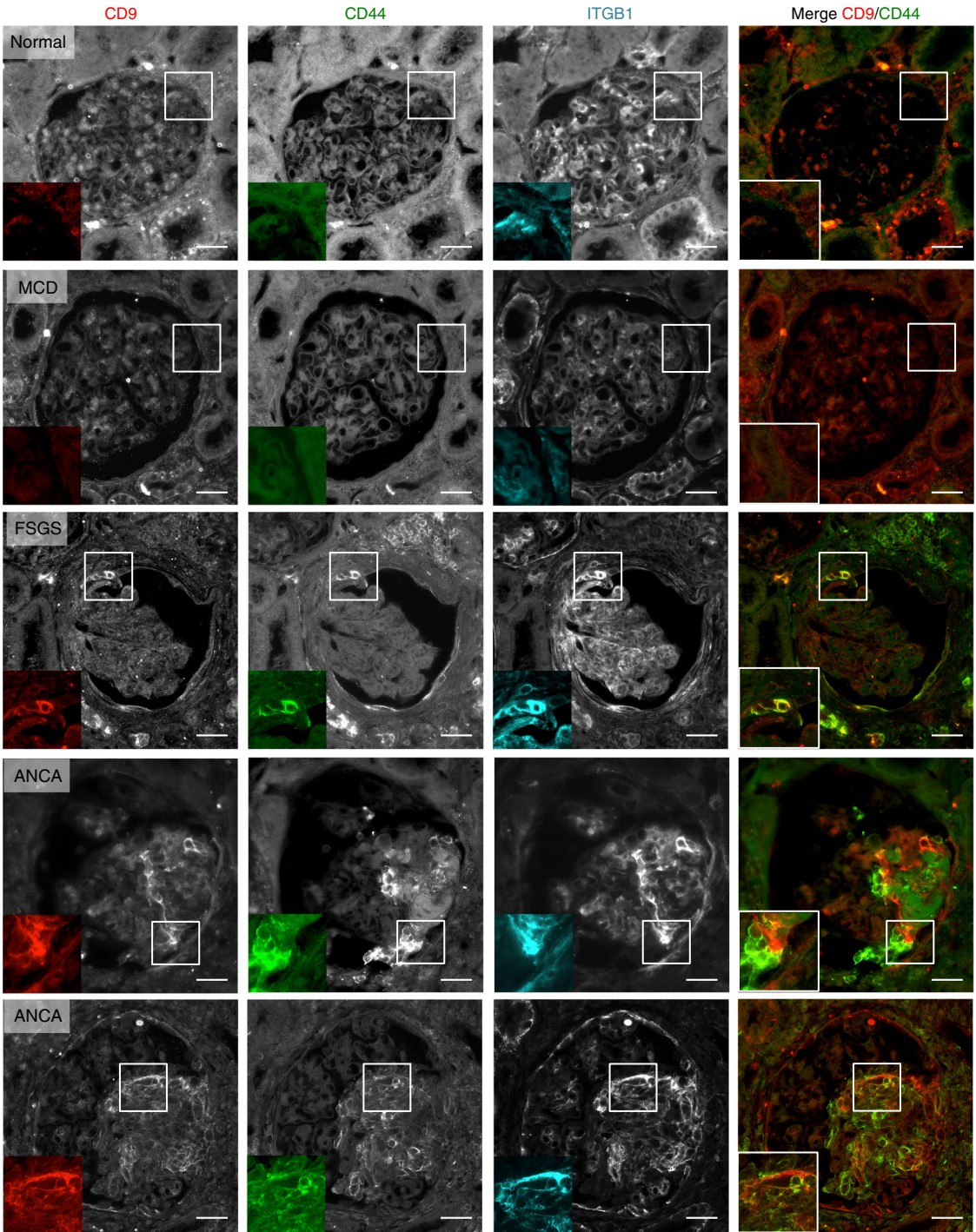

**Fig. 7** In human, CD9 overexpression in pathological glomeruli is associated with CD44 and ITGB1 expression. Representative images of immunofluorescent stainings of CD9 (left panel), CD44 (second panel), ITGB1/β1 integrin (third panel), and merge for CD9 (red) and CD44 (green) expression (right panel) on glomeruli from pathological human tissues. Scale bar, 50 μm. Higher magnifications are shown in the insets. MCD minimal change disease, FSGS focal segmental glomerulosclerosis, ANCA ANCA vasculitis. Source data are provided as a Source Data file

under the control of the selective PEC promoter (pPEC-rtTA-M2 mice) with mice expressing CRE recombinase under the control of an inducible promoter (tetracycline-responsive element). Doxycycline hyclate (Sigma Aldrich) was administered in drinking water for 14 days (5% sucrose and 1 mg/ml doxycycline) followed by 1 week of washout.

Age-matched littermates that had no deletion of *Cd9* in any cells were considered as controls. Animals were housed under standard specific pathogen-free conditions. All animal procedures were performed in accordance with guidelines of the European Community (L358–86/609EEC), and were approved by the Institut National de la Santé et de la Recherche Médicale and the Ministry for Higher Education and Research (MESR 7646 and TARGET GLOMDIS).

**Bone marrow transplantation**. Recipient mice were exposed to 9.5 Gy Cesium 137 radiation for 10 min. The day after, each mouse received $10 \times 10^6$ bone marrow cells by intravenous injection (i.v.) under isoflurane anesthesia. Mice received antibiotics treatment (Enrofloxacine 0.025%) in drinking water for 2 weeks. Chimeric mice were used 6 weeks after transplantation. Transplantation efficiency was determined to be > 90% in parallel experiments.

**Induction of nephrotoxic nephritis**. Nephrotoxic nephritis was induced on male mice (10–12 weeks of age) by i.v. of 15 μl of sheep anti-mouse glomerular basement membrane nephrotoxic serum, which was diluted with 85 μL of sterile PBS. Serum injections were repeated twice (on days 2 and 3)[66]. Mice were killed after 21 days.

**DOCA-salt and nephron reduction model**. In all, 10-to-15-week-old male mice underwent unilateral left nephrectomy via flank incision under isoflurane anesthesia and received 0.1 mg/kg buprenorphine twice daily for 2 days. They were divided into two groups receiving either deoxycorticosterone acetate (DOCA) or placebo. DOCA pellets (50 mg of DOCA per pellet) and placebo pellets with 21-day release (Innovative Research of America) were implanted subcutaneously 2 weeks after uninephrectomy. A second pellet of DOCA or placebo was implanted 3 weeks after the first implant. All animals received 0.9% NaCl in drinking water ad libitum. Mice were killed after 6 weeks of DOCA or placebo treatment.

**Blood pressure assessment**. Blood pressure was measured by using tail cuff plethysmography with the BP-2000 Blood Pressure Analyzing system (Visitech system). After a week of habituation, blood pressure was assessed weekly for 2 consecutive days during the entire experiment time course. Values represent the mean of the two measures.

**Biochemical measurements in blood and urine**. Urinary creatinine and BUN concentrations were analyzed by a standard colorimetric method (Olympus AU400) at the Biochemistry Laboratory of Institut Claude Bernard (IFR2, Faculté de Médecine Paris Diderot). Urinary albumin excretion was measured by using a specific ELISA for quantitative determination of albumin in mouse urine (Cell-Trend GmbH). Blood count was performed on a Hemavet counter (Drew scientific) on fresh blood after intracardiac puncture. Heparin was used as anticoagulant.

**Transmission electron microscopy procedure**. Small pieces of the renal cortex were fixed in Trump fixative (EMS) and embedded in Araldite M (Sigma Aldrich). Ultrathin sections were counterstained with uranyl acetate and lead citrate and then examined in a JEOL 1011 transmission electron microscope with Digital Micrograph software for acquisition.

**Human tissues**. Acetic acid-formol-alcohol-fixed, paraffin-embedded renal tissue specimens were obtained from the Pathology department of Hôpital Européen Georges Pompidou, Assistance Publique-Hôpitaux de Paris, Paris, France. Human tissue was used after informed consent by all the patients and approval form, and kidney biopsies collection was approved by the Inserm Ethics Committee (IRB00003888, FWA00005831 NIH OHRP, Office of Human Research Protection). Kidney biopsy specimens were collected in compliance with all relevant ethical regulations, and those with sufficient tissue for immunohistochemical evaluation after the completion of diagnosis workup were included.

**Histology**. Kidneys were immersed in 10% formalin and embedded in paraffin. Sections (4-μm thick) were processed for histopathology or immunohistochemistry. Sections were stained with Masson's trichrome, periodic-acid Schiff or red Sirius staining. For immunofluorescence, paraffin-embedded sections were stained with the following primary antibodies: rabbit anti-WT1 (Abcam, ab89901, 1:100), goat anti-PODXL (R&D systems, BAF1556, 1:200), rabbit anti-CD9 (Abcam, ab92726, 1:100), rat anti-CD44 (Abcam, ab119348, 1:100), rat anti-CD44 AF647-conjugated (Biolegend, 103018, 1:100), rabbit anti-integrin ß1 (Abcam, ab179471, 1:500), goat anti-synaptopodin antibody (Santa Cruz Biotechnology; SC21537; 1:400), rabbit anti-p57 antibody (Santa Cruz Biotechnology; SC8298; Santa Cruz; 1:200), goat anti-podoplanin (R&D Systems; AF3244-SP; 1:400), mouse anti-PCNA (Abcam; ab29; 1:400), and guinea pig anti-synaptopodin (Synaptic Systems; 163004; 1:500).

The following secondary antibodies were used: donkey anti-goat IgG (H + L) AF488-conjugated antibody (Life Technologies, A-11005, 1:400), donkey anti-rabbit IgG (H + L) AF594-conjugated antibody (A-21207, Life Technologies, 1:400), donkey anti-rat IgG (H + L) AF488-conjugated antibody (A-21208, Life Technologies, 1:400), donkey anti-rabbit IgG(H + L) Cy5-conjugated antibody (Jackson Immunoresearch, 1:200). The nuclei were stained with DAPI. Photomicrographs were taken with an Axiophot Zeiss photomicroscope (Jena, Germany).

For immunohistochemistry, paraffin-embedded sections were stained with primary mouse anti-CD9 antibody for human tissues (kindly provided by Claude Boucheix and Eric Rubinstein) or rabbit anti-CD9 (Abcam, ab92726, 1:100). Sections were then incubated with Histofine® (Nichirei Biosciences, Japan) during 2 h at room temperature. Staining was revealed with AEC reagent (DAKO), counterstained with hematoxylin QS (Vector, Burlingame, CA), and finalized with Permanent Aqueous Mounting Media (Innovex).

For quantification of histological lesions, an average of 40 glomerular cross-sections per mice was blindly counted for crescents in NTN model and abnormal glomeruli, sclerotic lesion above 50% of glomerulus surface and FSGS lesions in the DOCA-salt model. For red sirius quantification, automatic scanning of the whole kidney section was performed using Vectra technologies. Quantification of red Sirius area was then performed on 13 randomly chosen cortical fields per mice. Initially, podocyte number was assessed by the number of WT-1+ cells per glomerular cross-section in an average of 50 glomerular cross-sections per mice.

**Model-based stereology**. Podocyte depletion indices were assessed by the Weibel and Gomez method[67] validated by White and Bilous[68] in the kidney. This is one of the preferred methods to determine podocyte depletion parameters when limited tissue is available[69,70]. Briefly, 1-μm-thick paraffin sections were cut and stained with anti-p57, anti-SNP, and anti-CD44 antibodies and conjugated with AF488, AF594, and AF647, respectively. A minimum of ten glomerular cross-sections per mouse was systematically and randomly selected from the entire renal cortex. Images were obtained with a confocal laser microscope (LSM 710; Zeiss, Germany) running the Zen 2009 (Zeiss) software.

In order to estimate glomerular volume, podocyte number and thereby podocyte density (podocyte number divided by glomerular volume), we determined the number of podocyte nuclei per glomerular cross-section based on nuclear p57 and cytoplasmic SNP expression, total area of podocyte nuclei based on p57 expression, and glomerular tuft area. These parameters were inserted in the following formulae:

The number of podocytes per glomerulus ($N_{pod}$) was estimated by:

$$N_{pod} = N_v \times G_v$$

$N_V$ and $G_v$ represent podocyte density and glomerular volume, respectively. In order to obtain $N_v$, we used this equation:

$$N_v = \left(\frac{1}{\beta}\right) \times (N_{A^3} \div V_{v^{0.5}})$$

$N_A$ represents the division of the number of podocyte nuclei by the glomerular tuft area. $V_V$ gives a ratio of total podocyte nuclei area over glomerular tuft area. The β coefficient of 1.55 was assuming podocyte nuclei are ellipsoids. $G_v$ was calculated using this equation:

$$G_v = G_{A^{1.5}} \times (1.38 \div 1.01)$$

$G_A$ represents the glomerular tuft area, 1.38 is the sphere shape coefficient, and 1.01 is the size distribution coefficient that assumes a 10% coefficient of variation. Furthermore, total podocyte volume was obtained with the same equation:

$$P_v = P_{A^{1.5}} \times (1.38 \div 1.01)$$

$P_V$ represents total podocyte volume and $P_A$ the combination of nuclear (p57+) and cytoplasmic (SNP+) areas per glomerular cross-section. In order to obtain average podocyte volume, we divided total podocyte volume by total podocyte number.

During this process, glomeruli were also categorized based on the presence of CD44 + PEC, which were identified based on their anatomical location.

**PEC cell line**. Primary PEC line has been previously described[37] and were a kind gift from M Moeller. Primary PECs were cultured at 5% $CO_2$, and 37 °C in complete endothelial cell basal medium (PromoCell) with penicillin/streptomycin 1% (Thermo Fisher Scientific) and fetal bovine serum (FBS) 20% (Gibco) until 70% of confluence. The maintenance culture was passaged once a week by gentle trypsinization by using trysin EDTA 0.05% (Thermo Fisher Scientific).

**Cell silencing of Cd9 by small-hairpin RNA**. PECs at passage six were transduced with a lentivirus encoding for an anti-*Cd9* small-hairpin RNA (shRNA) (sh-*Cd9*) or a scramble shRNA (sh-scramble) at a dose of 20 multiplicity of infection (MOI). Medium containing lentiviral particles was replaced by fresh medium after 1 day. Cells were analyzed at least 7 days post transduction. Production of HIV1 delta U3 SIN lentiviral particles with VSV-G envelop was carried out by the VVTG facility platform (Necker faculty) by using the lentiviral vector TRC1-pLKO.1-U6- shR NA*Cd9* (MISSION shRNA, SHCLND NM_007657, TRCN0000066393, Sigma Aldrich) containing *Cd9*-specific shRNA (sequence: CCGGCCTGCAATGAAAGG TACTATACTCGAGTATAGTACCTTTCATTGCAGGTTTTTG) or the control vector (MISSION NON-TARGET SHRNA CONTROL VECTOR, SHC002, Sigma Aldrich).

**Cell adhesion assay and cell surface evaluation**. In total, 20,000 PEC were plated in six-well culture plates. Photomicrographs were taken at magnification ×1000 for 30 min, 1, 6, and 24 h after plating. The number of floating and adherent cells was assessed on 15 pictures (average number of cells counted per experiment, n = 460). Cell surface was evaluated at 24 h of plating on an average number of 100 cells per condition by using ImageJ software.

**Cell migration in IBIDI chambers**. Unoriented migration assay was performed using IBIDI chambers (Culture-Insert 2 Well in μ-Dish 35 mm, high ibiTreat, n° 81176). After confluence was obtained, PECs were grown in ECBM without supplements with FBS 1% overnight. The insert was then removed creating a gap of 500 μm between cells. A first round of photographs was taken (H0), shortly after cells were stimulated with either ECBM without supplements with FBS 1% (control), or with HB-EGF 10 ng/mL, or with PDGF-BB 10 ng/mL. Photographs were taken every 30 min for 12 h. Areas of migration were measured by automatic quantification on NIS-Elements AR software.

**Migration in PDGF gradient using microfluidic devices**. The manufacture of microchannels has been previously described[71]. Briefly, we use a soft lithography technique to obtain a mold with the desired channel architecture. Poly-dimethylsiloxane (PDMS) is poured over the mold and cured at 70 °C (Fig. 4e). Afterward, using plasma bonding, the T-shape PDMS microchannel is irreversibly attached to a glass coverslip. The width of the microchannel is 100 μm.

Parietal epithelial cells are cultured in the ECBM medium, 20% FBS, 1% penicillin streptomycin) at 37 °C. Channels are treated with plasma cleaner to render PDMS hydrophilic. FBS is injected into the channel 1 h before cell injection and incubated at 37 °C. After cells are dissociated with trypsin and resuspended in culture medium, centrifugation allows a high cell concentration (~$10^7$ cells/mL). Cells are injected into the channel and incubated at 37 °C during 24 h. The day after, cells are starved using ECBM + 1% FBS overnight. Cells are used for the experiment the next day. We typically perform experiments on multiple cells. The localization of the observed cells in the channel is chosen in order to maximize the gradient of growth factor. PDGF-BB is used at 20 ng/mL and HBEGF at 20 ng/mL. Growth factors are diluted in HBSS-HEPES 10 mM. Cells are observed under an Olympus IX3-CBH microscope using CellSens Dimension software. Figure 5g shows the geometry of the PDMS microchannel and depicts a typical experiment under PDGF-BB stimulation. Two syringe pump driven entries are used to supply liquid flow. Flow rate is 3 μL/min allowing low shear-stress forces inside the channel, to ensure low Reynold's number and laminar flow.

Every observable cell is analyzed at the beginning and after 3 h of observation. Cells were assessed for shape centroid coordinates at both time using ImageJ software. X axis is oriented in the flow, and Y axis along longitudinal displacement (i.e. along gradient of concentration). Difference of X and Y coordinates are plotted on the graphs. For every initial position of cells (Y), the gradient of concentration could be approximated using the following equation:

$$C(Y) = \frac{C0}{2} * erfc\frac{Y}{\sqrt{2.D.T}}$$

in which CO is the initial concentration of growth factor (CO = 20 ng/mL for PDGF-BB and 20 μmol/L for HB-EGF), erfc stands for complementary error function, Y is the centroid coordinate of the cell in the canal, D is the coefficient of diffusion of the growth factor and T the time necessary for the flow to reach the X coordinate of the cell. Only cells exposed to a $\frac{\Delta C}{C}$ > 100% were taken into account. This represent the cells localized in the centrum of the microchannel. Absolutes displacements were evaluated using the difference in X or Y between initial and final position.

**Cell-apoptosis assay**. Cells viability was assessed using APC conjugated Annexin-V (ImmunoTools) combined with propidium iodide (PI) (Thermo Fisher Scientific). Cells were plated in six-well plates during 24 h. Cells were then trypsinized and washed in cold phosphate-buffered saline (PBS w/o Ca$^{2+}$). Supernatants were kept and included in analysis. After centrifugation, cells were then resuspended in 100 μl of Annexin-V binding buffer (BD Pharmingen) and directly stained with Annexin-V and PI at room temperature for 5 min in the dark. A control tube without Annexin-V binding buffer was used to sustain aspecific signal from final analysis. Samples were analyzed on a BD LSRII flow cytometer (BD Bioscience). The data were analyzed using FlowJo software (FlowJo, LLC). Viable cells were defined as both Annexin-V and PI negative, apoptotic cells as Annexin-V positive and PI negative, necrotic cells as both Annexin-V and PI positive, and all populations were expressed as percentage of total cells.

**Proliferation assay**. To assess cell proliferation, we used the nuclear antigen Ki67 staining. After overnight starvation in ECBM + 1% FBS, cells were treated 24 h with PDGF-BB (10 ng/mL) or HBEGF (20 ng/mL) or corresponding controls. Cells were then trypsinized and washed in cold PBS. Supernatants were kept and included in analysis. After centrifugation, cells were permeabilized with 1 mL of ice cold ethanol for 2 h at −20 °C. Following two washes with FACS buffer (HBSS, 2% FBS, and 10 mM HEPES), cells were stained for 30 min at room temperature in the dark with phycoerythrin (PE)-conjugated anti-human Ki67 mAb (clone B56) (BD Pharmingen) and washed. Samples were analyzed on a BD LSRII flow cytometer (BD Bioscience). An isotype control staining with PE mouse IgG1 was performed. The data were analyzed using FlowJo software (FlowJo, LLC).

**Western blot analysis**. PEC lysates were prepared with RIPA extraction buffer containing phosphatase inhibitors and protease inhibitors (Roche). Equal amounts of proteins were loaded onto sodium dodecyl sulfate–polyacrylamide electrophoresis gels for separation and transferred onto poly(vinylidene difluoride) membranes. The membranes were blocked with milk and probed with different antibodies: rat anti-CD9 (BD Pharmingen, 553758, 1:1000), rabbit anti-phospho-PDGF receptor-ß Y1009 (Cell Signaling Technology, 4549; 1:1000), rabbit anti-PDGFRβ (Cell Signaling, 3169; 1:1000), rabbit anti-integrin ß1 (Millipore, 04-1109, 1:1000), rabbit anti-CASP3 (Cell Signaling Technology, 9662; 1:1000), rabbit anti-phospho EGFR Y1068 (Cell Signaling Technology, 2234; 1:1000), rabbit anti EGFR (Cell Signaling Technology, 2232; 1:1000), rabbit anti-phospho-FAK Y397 (Cell Signaling 3283; 1:1000), rabbit anti-FAK (Millipore, 06-543, 1:1000). Membranes were then probed with horseradish peroxidase-conjugated secondary antibodies

(Cell Signaling Technology, 7074 and 7076; 1:2000; Jackson Immunoresearch, 706-036-148; 1:1000) and bands were visualized by enhanced chemiluminescence (Clarity Western ECL substrate; Bio-Rad, 170–5061). A LAS-4000 imaging system (Fuji, LAS4000, Burlington, NJ, USA) was used to reveal bands and densitometric analysis was used for quantification. Uncropped blots are shown in Supplementary Figs 18–22.

**Quantitative RT-PCR**. The total RNA extraction of mouse glomeruli and lung tissue was performed using Qiazol (Qiagen), according to the manufacturer's recommendations. RNA reverse transcription was performed using the Quantitect Reverse Transcription kit (Qiagen) according to the manufacturer's protocol. The Maxima SYBR Green/Rox qPCR mix (Fermentas; Thermo Fisher Scientific) was used to amplify cDNA for 40 cycles on an ABI PRISM thermo cycler. The comparative method of relative quantification (2-DDCT) was used to calculate the expression level of each target gene, normalized to Ppia. The data are presented as the fold change in gene expression.

The oligonucleotide sequences are available on the Supplementary Information as Supplementary Table 3.

**Statistical analysis**. Data are expressed as means ± s.e.m. Statistical analyses were calculated by using GraphPad Prism software (GraphPad Software, La Jolla, CA). Comparisons between two groups were performed using t tests (95% confidence interval) with Welch correction whenever the variances were different between groups. Comparisons between multiple groups were performed using one-way ANOVA or two-ways ANOVA followed by Tukey post-test. Spearman rank coefficients were used in order to assess associations between two continuous variables. A P-value < 0.05 was considered significant.

**Reporting summary**. Further information on research design is available in the Nature Research Reporting Summary linked to this article.

## Data availability
The authors declare that the data supporting the findings of this study are available within the paper and its supplementary information files. Source data for Figs 1b, 2a, b, 3a, b, d, 4c, d, g–i, 5a b, d, h, i, and 6b, e–k and supplementary figures are provided as a Source Data file. All data are available from the corresponding authors upon reasonable request. A reporting summary for this article is available as a Supplementary Information file.

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

## Acknowledgements

This work was supported by Institut National de Santé et de la Recherche Médicale (INSERM), research grants from the European Research Council under the European Union's Seventh Framework Programme (FP7/2007-2013)/ ERC grant agreement n°107037), European Research Projects on Rare Diseases E-Rare-2 JTC 2011 from l'Agence Nationale de la Recherche (ANR) of France and the Freiburg Institute for Advanced Studies to P.-L.T. We thank National Health and Medical Research Council of Australia, the Humboldt Foundation and German Society of Nephrology for supporting VGP. We are grateful to Assistance Publique des Hôpitaux de Paris for supporting H.L. We thank the European Research Council for supporting C.H. and O.L. We thank Nano-K for supporting the Laboratoire d'Optique et de Biosciences at Ecole Polytechnique. We also thank Elizabeth Huc, Nicolas Perez, and the ERI970 team for assistance in animal care and handling, Nicolas Sorhaindo for biochemical measurements (ICB-IFR2, Laboratoire de Biochimie, Hôpital Bichat, Paris, France), Alain Schmitt and Jean-Marc Masse for transmission electron microscopy (Institut Cochin, Paris, France), Corinne Lesaffre for histological stainings, and technical support by Valerie Oberüber and Anja Obser. We acknowledge excellent administrative support from Véronique Oberweis, Annette De Rueda, Martine Autran, Bruno Pillard and Philippe Coudol. Dr P-L Tharaux was supported by the Freiburg Institute for Advanced Studies (FRIAS).

## Author contributions

H.L., C.H., O.L., Cédric B. and P.-L.T. wrote the paper. A.A., Cédric B. and P.-L.T. designed the study. C.H., O.L., V.G.P., S.F., E.C., N.D., L.M., M.J.M., T.B.H., E.R., F.L.N., C.l.B., A.A. and P.L.T. edited the paper. H.L., C.H., O.L., O.K., C.F., François G., F.l.G., V.G.P., Cédric B. and M.F. performed experiments and analyzed data. G.B., M.C., L.G., C.F., Corinne M., Chantal M., F.B., A.C., B.R. and O.K. performed experiments. D.N., P.B., E.T. and A.K. provided and analyzed human samples. Cl.B., E.R. and F.L.N. designed and provided *Cd9* flox mouse. E.L., C.H., O.L., and V.G.P. contributed equally to this article. A.K., F.L.N., E.R., C.l.B., A.A. and M.J.M. contributed equally to this article.
