## [Peer Review File · Nature Communications]

Reviewers' comments:

Reviewer #1 (Remarks to the Author):

Using various conditional knock out mouse lines, this study demonstrated that CD9 in PECs drives PEC activation, proliferation, and migration. This finding is consistent with CD9 expressions in CGN and FSGS mouse models and in human proliferative glomerulonephritis. They also reported that PEC-specific Cd9 depletion impairs cell migration and proliferation partly through EGFR and PDGFR pathways. Their discoveries are significant and novel in terms of understanding the pathogenesis of aforementioned glomerular diseases as well as CD9 pathophysiology.

There are several concerns listed below.

1. In Figure 5c-d, the readouts at different time points from the wound healing analysis (or migration in IBIDI chamber) of PECs cells are strongly recommended, instead of the report from only one time point (9 hour).
2. It is more important to examine the level of integrin beta 1 at the surface of PECs, in addition to the total level of cellular integrin beta 1 by Western blot (Figure 6a-b). Also, determine which beta 1 integrin(s) is(are) altered after CD9 knockdown in PECs will certainly provide important mechanistic insight to the story.
3. Analyses on EGFR/PDGFR signaling pathways are largely correlative in this study. Could manipulating EGFR/PDGFR pathways reduce the phenotypic differences between scrambled and CD9 shRNA groups? For example, could PECs with CD9 knockdown migrate better in microfluidic chamber in response to the stimulation of higher concentrations of PDGF and/or EGF? Also for EGFR/PDGFR signaling, it appears that CD9 knockdown affects EGFR and PDGFR levels at the first place, while reduced activations of EGFR/PDGFR signaling seem to be the secondary effects (Figure 6). Thus, the emphasis of this part of the study should be placed how CD9 knockdown affects the levels of EGFR and PDGFR.

Reviewer #2 (Remarks to the Author):

The manuscript by Lazareth et al focuses on CD9 in pathobiology of PECs in crescentic GN and FSGS. The rationale for studying CD9 followed its increase in a screen by RNA seq. The authors showed that although CD9 increased in both PECs and podocytes, only its deletion in PECs lead to changes in outcomes in 2 models of glomerular disease. Deletion in CD9 correlated with reduced PEC activation.

Deleting CD9 in cultured PECs reduced adhesion and PEC migration to PDGF-BB, and also reduced ITGB1 content. These results show a critical and novel role for CD9 in PEC biology in disease.

Comments:

1. The in vivo studies only show a correlation with CD9 and CD44. Did changes in CD9 levels in cultured PECs also reduce CD44 expression?
2. Did CD9 increase following uninephrectomy alone?
3. What is causing CD9 to increase in both PECs and podocytes?
4. Can the authors speculate why deleting CD9 in podocytes had no impact on outcomes?
5. The transgene used in the PEC mouse is expressed in both PECs and tubules, and is therefore not PEC-specific. Because both PECs and tubules will have CD9 deleted in the "PEC" transgenic, we need to know the expression of CD9 in tubules, given that tubule-interstitial fibrosis is improved

Reviewer #3 (Remarks to the Author):

In this study the authors performed the gene expression profiles in the glomeruli of NTS GN model and identified CD9 expression was highly increased at the early stage. Then, they confirmed that CD9 expression was increased mostly in PECs in both NTS and FSGS mouse and human kidneys. PEC-specific KO of CD9 attenuated glomerular cell injury (PEC and podocytes) and renal fibrosis while podocyte-specific KO of CD9 did not have any protective effects. They also confirmed that CD9 expression in platelet and BM-derived cells did not contribute to the progression of these diseases. In vitro, they found that CD9 knockdown reduced EGF and PDGF-induced PEC proliferation and migration. They also identified that CD9 may affect ITGB1 expression and cell adhesion.

Overall, the study was well done and the findings are novel and interesting. I have the following suggestions to improve the manuscript:

- 1) FSGS and crescentic GN have quite different presentations. The authors believe that CD9 has effects in both conditions. Could authors explain why CD9 induces PEC proliferation in crescent GN but PEC activation in FSGS?
- 2) What factors induce CD9 expression in GN or FSGS? It has been believed that podocyte injury may occur prior to the PEC activation. Does podocyte injury induce CD9 expression in PECs? In CD9 KO mice, it was a surprise for me that podocytes are completely normal. Does PEC activation injury podocytes?

- 3) The mechanism by which CD9 mediates PEC activation and proliferation is still unclear to the reviewer. Does CD9 function as a co-activator for all growth factors or does CD9 mediate integrin-induced pathway? What are the potential interacting partners of CD9 in PECs?
- 4) In the figure 1D, it would be better to include the co-staining of CD9 with a PEC cells-specific marker. Podocalyxin may also express in PECs. In Figure 1E, does the staining of CD9 in crescents localize only in PECs? Does CD9 staining also co-localize with inflammatory cell marker?
- 5) In the Figure 2C, the histologic picture does not reveal any typical crescent in NTS mice.
- 6) In the Figure 6C, does ITGB1 staining localize in both PECs and podocytes? Was there an increase of ITGB1 in podocytes too?
- 7) In the Figure 7, the staining of CD9 appeared to be increased only in a small area of glomerular tufts. In addition, search of Nephroseq.org did not reveal any increase of CD9 mRNA levels in any glomerular diseases. Therefore, the relevance of CD9 in human glomerular diseases requires further validation.

Dear Reviewers,

We have carefully considered all of your very helpful comments that we have addressed experimentally and discussed within the main text. New additions are highlighted in the manuscript text file.

On behalf of our co-authors, we would like to thank the reviewers for their constructive remarks on our manuscript. We hope we have answered their remarks and that the manuscript is now acceptable for publication in *Nature Communications*.

Reviewers' comments:

Reviewer #1 (Remarks to the Author):

Using various conditional knock out mouse lines, this study demonstrated that CD9 in PECs drives PEC activation, proliferation, and migration. This finding is consistent with CD9 expressions in CGN and FSGS mouse models and in human proliferative glomerulonephritis. They also reported that PEC-specific Cd9 depletion impairs cell migration and proliferation partly through EGFR and PDGFR pathways. Their discoveries are significant and novel in terms of understanding the pathogenesis of aforementioned glomerular diseases as well as CD9 pathophysiology.

There are several concerns listed below.

We appreciate these comments from the Reviewer and have now addressed these through a number of additional experiments.

1. In Figure 5c-d, the readouts at different time points from the wound healing analysis (or migration in IBIDI chamber) of PECs cells are strongly recommended, instead of the report from only one time point (9 hour).

We have reproduced the wound-healing assay with a continuous monitoring of cell migration for 24h. Movies are now added as supplemental information and associated quantification for the first 12h is shown in figure 5d in replacement of the previous 9-hour time point.

Source data for the whole 24h of migration are shown in the "source data file" and representative movies are added in supplemental information.

24h follow up of cell migration confirmed our previous results and shows that PEC cells deficient for CD9 migrate less than control PEC cells at all times, at basal and in response to both HB-EGF and PDGF-BB (n=4 chambers per condition; 2-way ANOVA shSCR vs. shCD9 over time all conditions, p<0.001).

2. It is more important to examine the level of integrin beta 1 at the surface of PECs, in addition to the total level of cellular integrin beta 1 by Western blot (Figure 6a-b).

Immunofluorescence for ITGB1 in PEC cells transduced with either a scramble shRNA, either a *Cd9* shRNA demonstrate mislocalization and lower expression of ITGB1 in cells deficient for CD9. Indeed, we observe that ITGB1 is localized at the cell membranes and in dots in the cytoplasm of scramble shRNA PECs; while ITGB1 expression level is low in *Cd9* shRNA PEC with diffuse staining and almost no membrane stained. Thus, CD9 seems to be responsible for ITGB1 stabilization and/or expression at the membranes.

ITGB1 immunofluorescence in PECs is shown in **Supplementary Figure 13** as shown below:

Also, determine which beta 1 integrin(s) is(are) altered after CD9 knockdown in PECs will certainly provide important mechanistic insight to the story.

We addressed this interesting questions and performed qPCR for *Itga* and *Itgb* mRNA expression in PECs transduced with a scramble shRNA or a *Cd9* shRNA. We found that mRNA expression of *Itgb-1* and *-3* and *Itga-1*, *-3*, *-10* and *-v* were decreased in CD9-deficient PEC while *Itga2* was induced. Interestingly, not only expression of these ECM receptors was decreased in CD9-deficient PEC but also epithelial and EMT markers mRNA expressions were modified. Indeed, we found that *Cldn1*, an epithelial tight junction differentiation marker of PEC, is strongly expressed in CD9-deficient PEC while less expressed in CD9+ cells. Conversely, EMT markers such as *Vimentin* and *Snail* were decreased in CD9-deficient PEC.

Thus, it appears that expression of CD9 in primary cultured PECs confers an “EMT-like” pro-fibrotic pro-migratory phenotype that could be reversed by shRNA-mediated CD9 knockdown.

RT-qPCR results are shown in **Supplementary figure 13** and discussion of these results has been added in the manuscript.

3. Analyses on EGFR/PDGFR signaling pathways are largely correlative in this study. Could manipulating EGFR/PDGFR pathways reduce the phenotypic differences between scrambled and CD9 shRNA groups? For example, could PECs with CD9 knockdown migrate better in microfluidic chamber in response to the stimulation of higher concentrations of PDGF and/or EGF?

We took advantage of the microfluidic system that enabled exposure of cells to finely tuned local concentrations of ligands within the micro-channel. We measured the mobility and migration of PECS with/without CD9 knockdown located off the microfluidic channel axis, which are thus exposed to higher PDGF concentration (≥ 2 -fold). No significant migration or increased mobility (**Supplementary figure 12**) compared to cells located in the vicinity of the channel axis (**Figure 4**) was observed. This observation indicates that a higher activation of PDGFR is insufficient to offset the CD9 knockdown-induced impairment of PEC migration and thus does not support the idea that CD9 inhibition would result in a threshold elevation in PDGFR activation that could be compensated either by PDGF excess or PDGFR over-expression. These results are now mentioned the manuscript main text and a **Supplementary figure** has been included.

Supplementary figure 12. Average speed (A) and quadratic speed (B) of wild type and CD9 knocked-down PECs submitted to a PDGF gradient in a microchannel, either at low average PDGF concentration (10 ng.ml⁻¹, respectively in red (n=30) and blue (n=52)) or high average PDGF concentration (20 ng.ml⁻¹, grey (n=27)). X is the channel axis direction and Y the orthogonal axis, along which the gradient is produced.

Also for EGFR/PDGFR signaling, it appears that CD9 knockdown affects EGFR and PDGFR levels at the first place, while reduced activations of EGFR/PDGFR signaling seem to be the secondary effects (Figure 6). Thus, the emphasis of this part of the study should be placed how CD9 knockdown affects the levels of EGFR and PDGFR.

We agree with the reviewer that CD9 deficiency affects EGFR and PDGFR protein total levels in cultured PEC. We have performed RT-qPCR assessment of *Egfr* and *Pdgfrb* genes expression and observed decreased mRNA expression for both in CD9 KD PECs, suggesting a global transcriptional imprinting effect of CD9 expression on PEC, involving EMT (as shown with modulation of *vimentin*, *snail* and *Cldn1* expression) and enhanced tyrosine kinase signaling. These additional results are shown in **Supplementary Figure 13** (see below) of the revised manuscript. The transcriptional regulation of PEC phenotype by CD9 remains to be elucidated.

Nonetheless, the change in the **net cellular content of the phosphorylated receptors** (as normalized to tubulin cellular abundance) is dynamic and net phosphorylation (especially for phospho EGFR (Y1068)) is lower in CD9 deficient cells upon acute stimulation. Likewise, PDGF and HB-EGF-dependent FAK phosphorylation is also dynamically altered in CD9-deficient PECs. This contrasts with the stable levels of the total forms of the PDGFR β , EGFR and FAK.

Thus, CD9 stimulates EGFR and PDGFR signaling at two levels, at the level of dynamic phosphorylation cascades and at the transcriptional level. These findings are now discussed in the manuscript.

Reviewer #2 (Remarks to the Author):

The manuscript by Lazareth et al focuses on CD9 in pathobiology of PECs in crescentic GN and FSGS. The rationale for studying CD9 followed its increase in a screen by RNA seq. The authors showed that although CD9 increased in both PECs and podocytes, only its deletion in PECs lead to changes in outcomes in 2 models of glomerular disease. Deletion in CD9 correlated with reduced PEC activation. Deleting CD9 in cultured PECs reduced adhesion and PEC migration to PDGF-BB, and also reduced ITGB1 content. These results show a critical and novel role for CD9 in PEC biology in disease.

We appreciate these comments from the Reviewer and have now addressed these through a number of additional experiments.

Please also note that our attention on CD9 was not only because its was found increased in a screen by RNA seq but also because of its very clear increase at the protein level both in mice and human diseased kidneys, known potentiating involvement in HB-EGF and EGFR signaling, and localization in human glomeruli with extracapillary lesions.

Comments:

1. The *in vivo* studies only show a correlation with CD9 and CD44. Did changes in CD9 levels in cultured PECs also reduce CD44 expression?

We acknowledge that *in vivo* experiments demonstrate a correlation between CD9 and CD44 expression with no evidence of a direct potential regulation by CD9 or CD44 on each other although the near absence of CD44 expression by CD9-deficient PECs suggests that CD9 may be an upstream requirement for acquisition of CD44.

We have performed RT-qPCR for *Cd44* expression in cultured CD9 competent vs. CD9-deficient PEC and found that the latter cells presented significantly decreased mRNA expression of *Cd44* by 50%. Data are shown in **Supplementary Figure 13** and discussed in the manuscript.

Taken together, these results support the hypothesis that CD9 could participate in PEC activation and subsequent CD44 expression occurring in CGN and FSGS.

2. Did CD9 increase following uninephrectomy alone?

As shown in **Figure 4b**, CD9 expression did not increase following uninephrectomy alone, i.e. in iPec-cd9^{wt/wt} wild-type mice 8 weeks after uninephrectomy and sub-cutaneous implantation of a placebo pellet. Thus, uninephrectomy was not sufficient to induce CD9 expression in PECs and did not induce features of FSGS.

3. What is causing CD9 to increase in both PECs and podocytes?

Following this interesting and sound question, we have conducted extensive *in vitro* experiments, manipulating pro-mitogenic, pro-motile and pro-fibrotic pathways in cultured PECs. Unfortunately, these experiments have provided only negative results thus far. There was no regulation of CD9 at protein and mRNA levels after acute pharmacological inhibition of JNK, STAT3 or EGFR for up to 48h. It may well be that a combination of multiple factors is involved in this critical pathogenic event. It is also likely that this question can't be adequately addressed in cell culture because proliferating PECs in culture do display an "activated" phenotype as found in tissue injuries, with already high level of CD9, EMT and proliferation. Thus, attempts to reverse CD9 expression may uncover other pathways than the ones that

would trigger CD9 *de novo* expression in differentiated CD9 negative cells. To our knowledge, such cellular model is not available yet.

Further studies will be needed to address this important question that is borderline to the scope of our present study.

Nonetheless, we have added the following sentences and reference **in the Discussion of the manuscript**: “Triggers for CD9 overexpression are unknown. Proliferating PECs in culture do display an activated phenotype as found in tissue injuries, with already high level of CD9, EMT and proliferation. Thus, attempts to reverse CD9 expression may uncover other pathways than the ones that would trigger CD9 *de novo* expression in CD9 negative cells. To our knowledge, such cellular model is not available yet. CD9 was found to be upregulated by mechanical stress in cultured immortalized podocytes whereas most of the tetraspanins remained unaffected ¹. Thus, further work would be useful to ascertain the influence of mechanical stimuli in CD9 regulation in PECs. At last, we have no answer to the even more complex question of *in vivo* induction of CD9 in PECs”.

4. Can the authors speculate why deleting CD9 in podocytes had no impact on outcomes?

We appreciate the point raised by the Reviewer.

To explain why deleting CD9 in podocytes had no impact on outcomes, we can speculate on two different hypotheses.

First, while CD9 *de novo* expression by PECs in RPGN and FSGS is clear and strong in human samples and rodent models, CD9 higher expression by podocytes in those contexts is less clear. CD9 was early identified in human kidney ^{2, 3} and recently in mouse podocytes ¹. In human biopsies, we found CD9 expression in crescents, in PECs and in podocytes in CGN and FSGS. In mice, CD9 is localized in crescents and in synechia where it colocalized with ITBG1 and partially with CD44 expression. Meanwhile, we detected only few CD9/SNP double positive cells, either because podocyte expressing CD9 were synaptopodin negative, either because podocyte only re-expressed CD9 at low level in rodent. Thus, deleting CD9 in podocytes in mice would have no effect because its expression is not strongly expressed in pathology.

CD9/CD44/SNP/DAPI co-stainings are now shown in **Figure 1e and Supplementary Figure 1**.

Second, CD9 is involved in a different tetraspanin web in podocyte than in PEC, with distinct cellular functions. Therefore, we have added the following sentences to the **Discussion section**: “Podocyte specific *Cd9* gene targeting did not alter the course of NTS-induced CGN. In CGN, both podocytes⁴ and then PECs⁵ participate to crescent formation. This involves initial podocyte proliferation^{6, 7}. Interestingly, whereas we found that CD9 expression controlled PEC proliferation and migration *in vitro*, we did not observe any influence of CD9 on proliferation and migration of primary podocytes from CD9 KO mice as well as from Npsh2-Cre *Cd9*^{lox/lox} mice. Such differences are consistent with cell specific functions of CD9. The biological roles for CD9, as described for other tetraspanins, are likely to be highly dependent of the cellular context and interacting partners⁸ that would differ in PECs and podocytes”.

Individual outgrowth area for n=3 mice per genotype.

5. The transgene used in the PEC mouse is expressed in both PECs and tubules, and is therefore not PEC-specific. Because both PECs and tubules will have CD9 deleted in the “PEC” transgenic, we need to know the expression of CD9 in tubules, given that tubule-interstitial fibrosis is improved.

We agree that the PEC-CRE is also present in scattered tubular cells a subpopulation of tubular cells that in young and healthy mice should be very small in the kidney (Berger et al. PNAS 2014). We analyzed the expression of the CRE recombinase-induced recombination with a ROSA (mTomato/mGFP) reporter and confirmed weak expression of the CRE in some tubules, accounting for less than 3% of tubules.

Furthermore, we observed CD9 overexpression in tubules in the NTN model and confirmed that CD9 expression in tubules is still present in the iPec-Cd9^{lox/lox} in this NTN context, thus confirming that tubular expression of the iPec-cre does not occur in the majority of tubular cells.

Reviewer #3 (Remarks to the Author):

In this study the authors performed the gene expression profiles in the glomeruli of NTS GN model and identified CD9 expression was highly increased at the early stage. Then, they confirmed that CD9 expression was increased mostly in PECs in both NTS and FSGS mouse and human kidneys. PEC-specific KO of CD9 attenuated glomerular cell injury (PEC and podocytes) and renal fibrosis while podocyte-specific KO of CD9 did not have any protective effects. They also confirmed that CD9 expression in platelet and BM-derived cells did not contribute to the progression of these diseases. *In vitro*, they found that CD9 knockdown reduced EGF and PDGF-induced PEC proliferation and migration. They also identified that CD9 may affect ITGB1 expression and cell adhesion.

Overall, the study was well done and the findings are novel and interesting. I have the following suggestions to improve the manuscript:

We thank the Reviewer for their recognition of our study as “novel and interesting”.

1) FSGS and crescentic GN have quite different presentations. The authors believe that CD9 has effects in both conditions. Could authors explain why CD9 induces PEC proliferation in crescent GN but PEC activation in FSGS?

The Reviewer is absolutely correct. Despite different etiologies, PEC phenotype shows similarities in advanced FSGS and crescentic GN. In both pathological contexts PECs undergo activation with migration and proliferation. Of note, in both contexts PECs showed *de novo* expression of CD44 and CD44 have been demonstrated recently to be essential for PEC activation and proliferation in both experimental FSGS and CGN (Eymael et al. *Kidney Int* 2018)⁹.

Here we showed that shRNA-mediated CD9 deficiency is associated to less proliferation of PECs suggesting that CD9 not only drives PEC migration but also drives PECs proliferation. Whether PEC proliferation and migration are independent to each other or not is actually not clear.

To support such hypothesis, we have analyzed PECs proliferation *in vivo* in the NTN and DOCA-salt models and we show PEC proliferation in both models in the control mice but not in CD9-deficient mice. PCNA immunofluorescence showing PEC proliferation in NTN and DOCA-salt models is now shown in **Supplementary figure 11 (below)**. Thus, CD9 seems to be required for both PEC migration and proliferation in pathology.

Thus, we postulate that CD9 *de novo* expression could be an early event in PEC activation, that could drive PECs to either migration, either proliferation, either both, depending to second messenger signaling itself depending on the pathological context.

2) What factors induce CD9 expression in GN or FSGS?

Following this interesting and sound question, we have conducted extensive *in vitro* experiments, manipulating pro-mitogenic, pro-motile and pro-fibrotic pathways. Unfortunately, these experiments have provided only negative results thus far. There was no regulation of CD9 at protein and mRNA levels after acute pharmacological inhibition of JNK, STAT3 or EGFR. It may well be that a combination of multiple factors is involved in this critical pathogenic event. It is also likely that this question can't be adequately addressed in cell culture because proliferating PECs in culture do display an "activated" phenotype as found in tissue injuries, with already high level of CD9, EMT and proliferation. Thus, attempts to reverse CD9 expression may uncover other pathways than the ones that would trigger CD9 *de novo* expression in CD9 negative cells. To our knowledge, such cellular model is not available yet. Also, we have no answer to the even more complex question of *in vivo* induction of CD9 in PECs.

Further studies will be needed to address this important question that is borderline to the scope of our present study.

Nonetheless, we have added the following sentences and reference **in the Discussion of the manuscript**: "Triggers for CD9 overexpression are unknown. Proliferating PECs in culture do display an activated phenotype as found in tissue injuries, with already high level of CD9, EMT and proliferation. Thus, attempts to reverse CD9 expression may uncover other pathways than the ones that would trigger CD9 *de novo* expression in CD9 negative cells. To our knowledge, such cellular model is not available yet. CD9 was found to be upregulated by mechanical stress in cultured immortalized podocytes whereas most of the tetraspanins remained unaffected ¹. Thus, further work would be useful to ascertain the influence of mechanical stimuli in CD9 regulation in PECs. At last, we have no answer to the even more complex question of *in vivo* induction of CD9 in PECs".

It has been believed that podocyte injury may occur prior to the PEC activation. Does podocyte injury induce CD9 expression in PECs?

We thank the reviewer for this relevant comment that we have addressed with several experiments:

Podocyte loss is not sufficient to trigger CD9 expression in PECs and FSGS.

We have added the following discussion: "It has been believed that podocyte injury may occur prior to the PEC activation. Therefore, we evaluated whether podocyte injury would induce CD9 expression in PECs. To this end, we assessed CD9 glomerular expression in a

model characterized with accentuated podocyte loss and glomerulosclerosis upon genetic targeting of podocyte autophagy that exacerbates diabetic nephropathy (Lenoir et al. Autophagy 2015). In that model, we demonstrated increased proteinuria and podocyte injury with foot process effacement and loss of differentiation markers in *Nphs.cre Atg5^{lox/lox}* diabetic mice but never observed FSGS nor synechiae¹⁰. We re-analyzed these kidneys looking for the PEC activation marker CD44 and for CD9 expression. We found no CD44 and no CD9 expression in PECs, neither in diabetic WT nor in diabetic *Nphs.cre Atg5^{lox/lox}* mice despite the latter group showed marked podocyte injury and loss. Thus, in this specific case, podocyte injury is not sufficient to induce neither PEC activation nor CD9 *de novo* expression.

Conversely, we recently observed experimental FSGS without primary podocyte insult but with manipulation of the endothelial Hif2a/Epas1 pathway using mouse genetics¹¹. Upon chronic angiotensin II infusion and high salt diet, *Cdh5-CRE Epas1^{lox/lox}* mice developed similar degree of podocyte injury to their wild type counterparts. Surprisingly, FSGS lesions with CD44+ and Fibronectin+ activated PECs were observed only in *Cdh5-CRE Epas1^{lox/lox}* hypertensive mice¹¹. We next analyzed CD9 expression in kidneys and found *de novo* CD9 expression in PECs in the hypertensive C57BL6/J *Cdh5-CRE Epas1^{lox/lox}* mice only. *De novo* high CD9 expression was almost exclusively associated with FSGS lesions and CD44 expression as shown in **Supplementary Figure 16**. Altogether, this finding suggests that endothelial derived mediators may contribute to CD9 induction in PECs and FSGS. Thus, we speculate that podocyte loss is not sufficient to trigger CD9 expression in PECs and FSGS. Other or additional pathological local environment factors may be responsible for CD9 *de novo* expression in PECs and subsequent FSGS development". These novel analyzes are now added in **Supplemental figure 16** and discussed in the manuscript.

Supplementary figure 16: CD9 expression in activated CD44+ PECs in experimental FSGS. (a-b) Representative images of immunofluorescent stainings for CD9 (red) and CD44 (green) in (a) WT (*Epas1^{lox/lox}*) and *Cdh5.cre Epas1^{lox/lox}* hypertensive mice and (b) WT (*Atg5^{lox/lox}*) and *Nphs2.cre Atg5^{lox/lox}* diabetic mice. N=6-8 mice per genotype. Scale bar: 50 μ m.

In CD9 KO mice, it was a surprise for me that podocytes are completely normal. Does PEC activation injury podocytes?

We thank the Reviewer for noticing this novel finding. It is a salient surprise of our study that NTN CD9 KO mice exhibited intact podocytes and in the iPEC-CD9lox/lox DOCA mice, podocyte number was partly preserved compared to the iPEC-CD9wt/wt DOCA mice (**Supplementary Figure 8**). Indeed, we observed a significant reduction in podocyte density without CD44+ PECs in the iPec-Cd9lox/lox mice in NTN and DOCA-salt models. The new findings indicate that podocyte loss without CD9 signaling in the PECs does not lead to extracapillary lesion formation and is blunted. Thus, when PEC are less or not activated, podocyte injury is less important. Consistently, expansion of the ITGB1 expressing PECs

correlated with loss of podocyte marker NPHS2 (**Supplementary Fig. 14**). Surprisingly, our study supports, for the first time to our knowledge, the hypothesis that PEC activation injury podocyte. This point has been added to the **Discussion** section.

3) The mechanism by which CD9 mediates PEC activation and proliferation is still unclear to the reviewer. Does CD9 function as a co-activator for all growth factors or does CD9 mediate integrin-induced pathway? What are the potential interacting partners of CD9 in PECs?

With the Reviewer, we realize that intricate signaling of RTK and integrins is difficult (if not impossible) to dissect since, as it is well documented in the literature and extensively discussed in the manuscript, tetraspanins act as molecular scaffolds, involving multimolecular complexes of signaling molecules. CD9 may thus promote growth factor signaling through the formation and stabilization of tetraspanin-enriched signaling microdomains that promote HB-EGF/EGFR and PDGFR β receptor insertion in the plasma cell membrane as well as the β 1 integrin that has been found to be laterally associated with CD9 by co-IP experiments as referenced. This may actually be very relevant the pathophysiology as increased expression of both β 1 and β 3 integrins have been reported in human CGN^{12, 13}. In rodent models, β 1 integrin is overexpressed 7 days after CGN induction, promoting cell adhesion to the matrix proteins¹⁴, as discussed. The known interdependence of RTK and integrin signalings, as shown for example in cultured PECs where CD9 status influenced FAK signaling downstream EGFR and PDGFR β stimulation places CD9 at a crucial integrating hub.

In keeping with this interesting question, we wondered whether CD9 expression would display a more profound and sustained influence on PECs phenotype. Thus, in order to evaluate potential global CD9-dependent changes in PECs, we performed RT-qPCR for *gfr*, *Pdgfrb*, *Itga* and *Itgb* mRNA expression in PECs transduced with a scramble shRNA or a *Cd9* shRNA. We found that mRNA expression of *Itgb*-1 and -3 and *Itga*-1, -3, -10 and -v were decreased in CD9-deficient PEC while *Itga*2 was induced. Interestingly, not only expression of these ECM receptors was decreased in CD9-deficient PEC but also epithelial and EMT markers mRNA expressions were modified. Indeed, we found that *Cldn1*, an epithelial tight junction differentiation marker of PEC, is strongly expressed in CD9-deficient PEC while less expressed in CD9+ cells. Conversely, EMT markers such as Vimentin and Snail were decreased in CD9-deficient PEC. Thus, it appears that expression of CD9 in primary cultured

PECs confers an “EMT-like” pro-fibrotic pro-migratory phenotype that could be reversed by shRNA-mediated CD9 knockdown.

In addition, CD9 deficiency affects EGFR and PDGFR protein total levels in cultured PEC. RT-qPCR assessment of *Egfr* and *Pdgfrb* genes expression uncovered decreased mRNA expression for both in CD9 KD PECs, suggesting a global transcriptional imprinting effect of CD9 expression on PEC, involving EMT (as shown with modulation of *vimentin*, *snail* and *Cldn1* expression) and enhanced tyrosine kinase signaling. These additional results are shown in **Supplementary Figure 13** of the revised manuscript. The transcriptional regulation of PEC phenotype by CD9 remains to be elucidated.

Supplementary figure 13: *In vitro* CD9 deficiency in PEC modifies ITGB1 expression. (a) Representative images of immunofluorescent stainings for ITGB1 (green) in PEC M12 cells transduced with a scramble or a *Cd9* shRNA. Nuclei were counterstained with DAPI (blue). (b,c) Fold change in mRNA expression for (b) *Itgb1*, *Itgb3*, *Itgb5*, *Itga1* to *Itga11* and *Itgav* and (c) *Cd9*, *Cd44*, *Cldn1*, *vimentin*, *snail*, *Egfr* and *Pdgfrb* in PEC M12 cells transduced with a scramble shRNA (blue dots) or a *Cd9* shRNA (red dots). N= 4-5, each dot represent one cell culture well. * p<0.05, ** p<0.01

Nonetheless, as added in the **Discussion section**, “CD9 expression influenced the dynamics of the change in the net cellular content of the phosphorylated receptors (as normalized to tubulin cellular abundance) with lower net phosphorylation (especially for phospho EGFR (Y1068) in CD9 deficient cells upon acute stimulation. Likewise, PDGF and HB-EGF-dependent FAK phosphorylation is also dynamically altered in CD9-deficient PECs. This contrasts with the stable levels of the total forms of the PDGFR β , EGFR and FAK. Thus, CD9 stimulates EGFR and PDGFR signaling at two levels, at the level of dynamic phosphorylation cascades and at the transcriptional level.”

4) In the figure 1D, it would be better to include the co-staining of CD9 with a PEC cells-specific marker. Podocalyxin may also express in PECs. In Figure 1E, does the staining of CD9 in crescents localize only in PECs? Does CD9 staining also co-localize with inflammatory cell marker?

We followed this useful advice and have added a CD9/CD44/SNP co-staining in new **Figure 1e** and CD9/PDPN co-staining in **Supplementary Figure 2** that confirms PEC expression of CD9 in NTN-induced CGN.

In **Figure 1f**, in human CGN we found CD9 expression in injured glomeruli in endothelial cells, PECs and podocytes and in crescentic lesions (where parietal or podocyte origin of crescentic cells have not been identified here). CD9 expression was also found in inflammatory cells surrounding the Bowman's capsule, both in human pathologies (**Figure 1f**) and mouse models (**Figure 1, 3, 4 and Supplementary figure 2, 3**).

5) In the Figure 2C, the histologic picture does not reveal any typical crescent in NTS mice.

The Reviewer is absolutely correct. The picture was not well chosen and has been changed.

6) In the Figure 6C, does ITGB1 staining localize in both PECs and podocytes? Was there an increase of ITGB1 in podocytes too?

To answer the reviewer's question, we have performed double IF for ITGB1 and podocin and for ITGB1 and CD44 in NTN and DOCA-salt models. While ITGB1 is mainly and strongly expressed in basal membranes of podocytes and mesangial cells in WT control mice, ITGB1 becomes strongly expressed by activated PECs (CD44+) and also some rare podocytes (NPHS2+) in both pathologies. No such ITGB1 expression is found in iPec-Cd9^{lox/lox} NTN or iPec-Cd9^{lox/lox} DOCA mice.

These results are now shown in **Supplementary figure 13 and 14**.

7) In the Figure 7, the staining of CD9 appeared to be increased only in a small area of glomerular tufts. In addition, search of [Nephroseq.org](https://www.nephroseq.org) did not reveal any increase of CD9 mRNA levels in any glomerular diseases. Therefore, the relevance of CD9 in human glomerular diseases requires further validation.

We have analyzed other renal biopsies and used two different CD9 antibodies (Abcam and home-made mAb from Claude Boucheix) to confirm our results. We now have 49 renal biopsies that have been analyzed confirming CD9 *de novo* expression in CGN and FSGS. Data on the patients and CD9 renal expression are resumed in **Supplementary Table 1** and **Supplementary Table 2**.

Supplementary Table 1 : Human kidney biopsies : diagnosis, patient characteristics and CD9-ITGB1 staining

Gender	Age	Diagnosis	Relapse	Glomerular CD9 expression	Colocalization CD9 / ITGB1
Female	45	Normal kidney		-	-
Male	28	Normal kidney transplant		-	-
Male	61	Minimal change disease	First episode	-	-
Male	62	Minimal change disease	First episode	-	-
Female	44	Focal Segmental Glomerulosclerosis	First episode	+	+
Female	51	FSGS- Sickle Cell disease	First episode	+	+
Male	35	IgA nephropathy with FSGS	First episode	+	+
Male	31	Focal Segmental Glomerulosclerosis	First episode	+	+

Female	29	Collapsing FSGS – fibrous kidney	Relapse	-	-
Male	43	Granulomatosis with polyangiitis	First episode	+	+
Female	44	Granulomatosis with polyangiitis	First episode	+	+
Male	48	Granulomatosis with polyangiitis	First episode	+	+
Female	83	Micro-polyangiitis ANCA negative	Relapse	+	+
Male	54	Micro-polyangiitis MPO-ANCA-positive	First episode	+	+
Male	80	Micro-polyangiitis MPO-ANCA-positive	First episode	+	+
Female	73	Micro-polyangiitis MPO-ANCA-positive	Relapse	+	+
Female	75	Micro-polyangiitis MPO-ANCA-positive	Relapse	+	+
Female	65	Micro-polyangiitis PR3-ANCA-positive	First episode	+	+

Supplementary Table 2: Additional human kidney biopsies: diagnosis, patient characteristics and CD9-staining

Gender	Age	Diagnosis	Relapse	Glomerular CD9 expression	CD9 cellular expression in glomeruli
F	24	FSGS	First episode	+	PEC – podocytes
F	24	FSGS	Relapse	+	PEC and flocculocapsular synechia
F	28	FSGS	First episode	+	All PEC – podocytes
F	39	FSGS	First episode	+	
M	35	FSGS	First episode	+	Podocytes
F	54	HIVAN	First episode	+	
M	39	IgA nephropathy	First episode	+	Crescent
M	48	IgA nephropathy	First episode	+	Crescent – podocytes – PEC
F	20	IgA nephropathy	First episode	+	Podocytes – PEC – flocculocapsular synechia
M	43	IgA nephropathy	First episode	+	PEC – flocculocapsular synechia
M	55	IgA nephropathy	First episode	-	-
M	24	IgA nephropathy	First episode	-	-
F	33	IgA nephropathy	First episode	+	
M	30	IgA nephropathy	First episode	+	Sclerotic glomerulus
M	40	IgA nephropathy with secondary FSGS	First episode	+	Podocytes – PEC – flocculocapsular synechia
M	33	IgA nephropathy with secondary FSGS	First episode	+	Normal PEC
M	27	IgA nephropathy with secondary FSGS	First episode	+	Podocytes – PEC – flocculocapsular synechia
M	61	IgA nephropathy with secondary FSGS	First episode	+	Flocculocapsular synechia
F	35	IgA nephropathy with secondary FSGS	First episode	+	Sclerotic glomeruli
M	27	IgA nephropathy with secondary FSGS	First episode	+	Podocytes
F	26	Class IV Lupus nephritis	First episode	+	Podocytes – PEC
M	69	ANCA-negative vasculitis with secondary FSGS	Relapse	+	Crescent – flocculocapsular synechia
F	41	ANCA-negative vasculitis	First episode	+	Crescent – podocytes – PEC
F	59	ANCA-positive vasculitis	Relapse	+	Crescent – podocytes – PEC
F	67	MPO-ANCA-positive vasculitis	First episode	+	Crescent – podocytes – PEC

M	94	MPO-ANCA-positive vasculitis	First episode		Crescent – podocytes
F	53	MPO-ANCA-positive vasculitis	First episode	+	Crescent – PEC – podocytes
M	30	PR3-ANCA-positive vasculitis	First episode	+	Crescent – podocytes – PEC and floculocapsular synechia
M	49	PR3-ANCA-positive vasculitis	First episode	+	Crescent – PEC – podocytes
F	24	PR3-ANCA-positive vasculitis with secondary FSGS	Relapse	+	PEC – podocytes and floculocapsular synechia
M	46	PR3-ANCA-positive vasculitis	First episode	+	Crescent – PEC – podocytes and floculocapsular synechia

Furthermore, additional analyses on Nephroseq revealed several associations between increased CD9 mRNA expression and some glomerular diseases or rodent models of renal disease. Interestingly, when making comparison between CD9, CD44 and ITGB1 expression in that cohorts, we note that usually mRNA expression of these 3 genes evolve in the same way. Thus, transcriptomic data support increased Cd9 mRNA expression in a number of conditions. Meanwhile, given the magnitude of increased in CD9 protein abundance in extracapillary diseases, we don't rule out that post-translational upregulation of CD9 may have a significant contribution too.

Comparison of CD44, CD9 and ITGB1 in Hodgkin FSGS Glom
Over-expression in Group: Focal Segmental Glomerulosclerosis vs. Normal Kidney
log2 median-centered intensity

Legend

1. Normal Kidney (9) 2. Focal Segmental Glomerulosclerosis (8)

Hodgkin FSGS Glom

Am J Pathol 2010/10/01 30 samples
microarray Human
Affymetrix Human X3P Array 19,139 measured genes
Glomeruli

Note: Colors are z-score normalized to depict relative values within rows. They cannot be used to compare values between rows.

© 2018 The Regents of The University of Michigan. Images from Nephroseq may be used in publications with proper citation. The citation is as follows: Nephroseq (The Regents of The University of Michigan, Ann Arbor, MI) was used for analysis and visualization. For further information, refer to the terms of use.
Nephroseq Source:
[https://nephroseq.org/resource/main.html#fa:1N10539;ac:1N10539;1N10945;d:1N156636781;dso:geneOverex;dt:predefinedClass;ec:\[1N2\];epv:1N1.1N3;et:over:f:133670107;g:928;gt:boxplot;p:1N200015062;pg:1;pvf:8217,8248,16107;scr:datasets;ss:analysis;v:17](https://nephroseq.org/resource/main.html#fa:1N10539;ac:1N10539;1N10945;d:1N156636781;dso:geneOverex;dt:predefinedClass;ec:[1N2];epv:1N1.1N3;et:over:f:133670107;g:928;gt:boxplot;p:1N200015062;pg:1;pvf:8217,8248,16107;scr:datasets;ss:analysis;v:17)

Comparison of CD44, CD9 and ITGB1 in Nakagawa CKD Kidney

Over-expression in Discovery Set Group: Chronic Kidney Disease vs. Normal Kidney
log2 median-centered intensity

Rank	P-value	Fold Change	Gene	Reporter	Gene
2308	2.08E-12	2.05	ITGB1	A_32_P95397	ITGB1
8192	2.38E-5	2.90	CD44	A_24_P919538	CD44
12511	0.026	1.29	CD9	A_23_P76364	CD9

Legend

1. Normal Kidney (5) 2. Chronic Kidney Disease (48)

Nakagawa CKD Kidney

PLoS One 2015/08/28 61 samples
microarray Human
Agilent Whole Human Genome Microarray 4x44K (Probe Name Version) 19,063 measured genes

Kidney
Least Expressed Most Expressed
[Color scale from blue to red] Not measured

Note: Colors are z-score normalized to depict relative values within rows. They cannot be used to compare values between rows.

© 2018 The Regents of The University of Michigan. Images from Nephroseq may be used in publications with proper citation. The citation is as follows: Nephroseq (The Regents of The University of Michigan, Ann Arbor, MI) was used for analysis and visualization. For further information, refer to the terms of use.
Nephroseq Source: [https://nephroseq.org/resource/main.html#fa:1N10846;ac:1N10539.1N10945;d:1N156636784;dso:geneOverex;dt:predefinedClass;ec:\[1N2\];epv:1N1.1N3;et:over.f.195109487;g:928;gt:boxplot.p.1N200015275;pg:1.pvf.8217.8248.16107;scr:datasets;ss:analysis;v:17](https://nephroseq.org/resource/main.html#fa:1N10846;ac:1N10539.1N10945;d:1N156636784;dso:geneOverex;dt:predefinedClass;ec:[1N2];epv:1N1.1N3;et:over.f.195109487;g:928;gt:boxplot.p.1N200015275;pg:1.pvf.8217.8248.16107;scr:datasets;ss:analysis;v:17)

Comparison of CD44, CD9 and ITGB1 in Nakagawa CKD Kidney

Over-expression in Validation Set Group: Chronic Kidney Disease vs. Normal Kidney
log2 median-centered intensity

Rank	P-value	Fold Change	Gene	Reporter	Gene
1486	0.002	2.83	ITGB1	A_32_P95397	ITGB1
1512	0.002	5.71	CD44	A_24_P919538	CD44
7012	0.021	2.43	CD9	A_23_P76364	CD9

Legend

1. Normal Kidney (3) 2. Chronic Kidney Disease (5)

Nakagawa CKD Kidney

PLoS One 2015/08/28 61 samples
microarray Human
Agilent Whole Human Genome Microarray 4x44K (Probe Name Version) 19,063 measured genes

Kidney
Least Expressed Most Expressed
[Color scale from blue to red] Not measured

Note: Colors are z-score normalized to depict relative values within rows. They cannot be used to compare values between rows.

© 2018 The Regents of The University of Michigan. Images from Nephroseq may be used in publications with proper citation. The citation is as follows: Nephroseq (The Regents of The University of Michigan, Ann Arbor, MI) was used for analysis and visualization. For further information, refer to the terms of use.
Nephroseq Source: [https://nephroseq.org/resource/main.html#fa:1N10847;ac:1N10539.1N10945;d:1N156636784;dso:geneOverex;dt:predefinedClass;ec:\[1N2\];epv:1N1.1N3;et:over.f.195109487;g:928;gt:boxplot.p.1N200015276;pg:1.pvf.8217.8248.16107;scr:datasets;ss:analysis;v:17](https://nephroseq.org/resource/main.html#fa:1N10847;ac:1N10539.1N10945;d:1N156636784;dso:geneOverex;dt:predefinedClass;ec:[1N2];epv:1N1.1N3;et:over.f.195109487;g:928;gt:boxplot.p.1N200015276;pg:1.pvf.8217.8248.16107;scr:datasets;ss:analysis;v:17)

Comparison of CD44, CD9 and ITGB1 in Peterson Lupus Glom

Over-expression in Group: Lupus Nephritis vs. Normal Kidney
log2 median-centered ratio

Rank	P-value	Fold Change	Gene	Reporter	Gene
66	1.53E-5	1.54	CD44	1709	CD44
503	0.028	1.23	CD9	3193	CD9
1628	0.714	-1.08	ITGB1	2881	ITGB1

Legend

1. Normal Kidney (6) 2. Lupus Nephritis (25)

Peterson Lupus Glom

J Clin Invest 2004/06/01
31 samples
microarray
Human
Custom platform
2,620 measured genes
Glomeruli

Note: Colors are z-score normalized to depict relative values within rows. They cannot be used to compare values between rows.

© 2018 The Regents of The University of Michigan. Images from Nephroseq may be used in publications with proper citation. The citation is as follows: Nephroseq (The Regents of The University of Michigan, Ann Arbor, MI) was used for analysis and visualization. For further information, refer to the terms of use.
Nephroseq Source: [https://nephroseq.org/resource/main.html#fa:1N10581.ac:1N10539.1N10945.d:1N156636786;dso:geneOverex.d:predefinedClass.ec:\[1N2\]epv:1N1.1N3.et:over.f:139931897.g:928.gt:boxplot.p:1N200015116.pg:1.pvf:8217.8248.16107.scr:datasets.ss:analysis.v:17](https://nephroseq.org/resource/main.html#fa:1N10581.ac:1N10539.1N10945.d:1N156636786;dso:geneOverex.d:predefinedClass.ec:[1N2]epv:1N1.1N3.et:over.f:139931897.g:928.gt:boxplot.p:1N200015116.pg:1.pvf:8217.8248.16107.scr:datasets.ss:analysis.v:17)

Comparison of CD44, CD9 and ITGB1 in Neusser Hypertension Glom

Over-expression in Group: Nephrosclerosis vs. Tumor Nephrectomy
log2 median-centered intensity

Rank	P-value	Fold Change	Gene	Reporter	Gene
349	4.97E-5	1.15	ITGB1	216178_x_at	ITGB1
1740	0.004	1.23	CD44	204490_s_at	CD44
2842	0.017	1.47	CD9	201005_at	CD9

Legend

1. Tumor Nephrectomy (4) 2. Nephrosclerosis (14)

Neusser Hypertension Glom

Am J Pathol 2010/02/01
18 samples
microarray
Human
Affymetrix Human Genome U133A Array
12,624 measured genes
Glomeruli

Note: Colors are z-score normalized to depict relative values within rows. They cannot be used to compare values between rows.

© 2018 The Regents of The University of Michigan. Images from Nephroseq may be used in publications with proper citation. The citation is as follows: Nephroseq (The Regents of The University of Michigan, Ann Arbor, MI) was used for analysis and visualization. For further information, refer to the terms of use.
Nephroseq Source: [https://nephroseq.org/resource/main.html#fa:1N10571.ac:1N10539.1N10945.d:1N156636785;dso:geneOverex.d:predefinedClass.ec:\[1N2\]epv:1N1.1N3.et:over.f:536798.g:928.gt:boxplot.p:1N200015105.pg:1.pvf:8217.8248.16107.scr:datasets.ss:analysis.v:17](https://nephroseq.org/resource/main.html#fa:1N10571.ac:1N10539.1N10945.d:1N156636785;dso:geneOverex.d:predefinedClass.ec:[1N2]epv:1N1.1N3.et:over.f:536798.g:928.gt:boxplot.p:1N200015105.pg:1.pvf:8217.8248.16107.scr:datasets.ss:analysis.v:17)

Comparison of CD44, CD9 and ITGB1 in Ju CKD Glom
 Over-expression in Group: Focal Segmental Glomerulosclerosis vs. Healthy Living Donor
 log2 median-centered intensity

Rank	P-value	Fold Change	Gene	Reporter	Gene
326	2.61E-6	1.47	CD44	960	CD44
2121	0.012	1.10	CD9	928	CD9
6356	0.829	-1.02	ITGB1	3688	ITGB1

Legend

1. Healthy Living Donor (21) 2. Focal Segmental Glomerulosclerosis (25)

Ju CKD Glom

Genome Res 2013/11/01 199 samples
 microarray Human
 Affymetrix Human Genome U133 Plus 2.0 Array (allCDF v10) 17,379 measured genes

Note: Colors are z-score normalized to depict relative values within rows. They cannot be used to compare values between rows.

© 2018 The Regents of The University of Michigan. Images from Nephroseq may be used in publications with proper citation. The citation is as follows: Nephroseq (The Regents of The University of Michigan, Ann Arbor, MI) was used for analysis and visualization. For further information, refer to the terms of use.
 Nephroseq Source: [https://nephroseq.org/resource/main.html#fa:1N10828;ac:1N10539;1N10945;d:1N156636782;dso:geneOverex;dt:predefinedClass;ec:\[1N2\];epv:1N1.1N3;et:over.f.195889137;g:928;gt:boxplot.p;1N200015073;pg:1.pvf.8217.8248.16107;scr:datasets;ss:analysis;v:17](https://nephroseq.org/resource/main.html#fa:1N10828;ac:1N10539;1N10945;d:1N156636782;dso:geneOverex;dt:predefinedClass;ec:[1N2];epv:1N1.1N3;et:over.f.195889137;g:928;gt:boxplot.p;1N200015073;pg:1.pvf.8217.8248.16107;scr:datasets;ss:analysis;v:17)

Comparison of CD44, CD9 and ITGB1 in Hodgkin Diabetes Mouse Glom
 Over-expression in eNOS-deficient C57BLKS db/db Group: Diabetic Nephropathy Mouse Model vs. Non-Diabetic Mouse Kidney

Rank	P-value	Fold Change	Gene	Reporter	Gene
299	3.18E-4	1.47	CD9	12527	CD9
351	4.51E-4	2.87	CD44	12505	CD44
7452	0.660	-1.04	ITGB1	16412	ITGB1

Legend

1. Non-Diabetic Mouse Kidney (5) 2. Diabetic Nephropathy Mouse Model (7)

Hodgin Diabetes Mouse Glom

Diabetes 2013/01/01 39 samples
 microarray Mouse
 Custom platform 14,789 measured genes

Note: Colors are z-score normalized to depict relative values within rows. They cannot be used to compare values between rows.

© 2018 The Regents of The University of Michigan. Images from Nephroseq may be used in publications with proper citation. The citation is as follows: Nephroseq (The Regents of The University of Michigan, Ann Arbor, MI) was used for analysis and visualization. For further information, refer to the terms of use.
 Nephroseq Source: [https://nephroseq.org/resource/main.html#fa:1N10824;ac:1N10539;1N10945;d:1N156636780;dso:geneOverex;dt:predefinedClass;ec:\[1N2\];epv:1N1.1N3;et:over.f.186292277;g:928;gt:boxplot.p;1N200015252;pg:1.pvf.8217.8248.16107;scr:datasets;ss:analysis;v:17](https://nephroseq.org/resource/main.html#fa:1N10824;ac:1N10539;1N10945;d:1N156636780;dso:geneOverex;dt:predefinedClass;ec:[1N2];epv:1N1.1N3;et:over.f.186292277;g:928;gt:boxplot.p;1N200015252;pg:1.pvf.8217.8248.16107;scr:datasets;ss:analysis;v:17)

Comparison of CD44, CD9 and ITGB1 in Berthier Lupus
 Mouse Kidney
 Over-expression in NZW/BXSB Model: Proteinuria (Mouse) 13 - 20 Weeks vs. No Proteinuria (Mouse) 5 - 8 Weeks

Rank	P-value	Fold Change	Gene	Reporter	Gene
7	3.48E-6	2.99	CD44	12505	CD44
445	0.002	1.29	CD9	12527	CD9
675	0.005	1.19	ITGB1	16412	ITGB1

Legend

1. No Proteinuria (Mouse) 5 - 8 Weeks (4) 2. Proteinuria (Mouse) 13 - 20 Weeks (6)

Berthier Lupus Mouse Kidney

J Immunol 2012/07/15 68 samples
 microarray Mouse
 Custom platform 14,789 measured genes
 Kidney

Note: Colors are z-score normalized to depict relative values within rows. They cannot be used to compare values between rows.

© 2018 The Regents of The University of Michigan. Images from Nephroseq may be used in publications with proper citation. The citation is as follows: Nephroseq (The Regents of The University of Michigan, Ann Arbor, MI) was used for analysis and visualization. For further information, refer to the terms of use.
 Nephroseq Source: [https://nephroseq.org/resource/main.htm#fa:1N10971.ac:1N10539.1N10945.d:1N156636775;dso:geneOverex:dt:predefinedClass;ec:\[1N2\].epv:1N1.1N3;et:over:f:186292277;g:928;gt:boxplot;p:1N200015490;pg:1;pvf:8217.8248.16107;scr:datasets:ss:analysis:v:17](https://nephroseq.org/resource/main.htm#fa:1N10971.ac:1N10539.1N10945.d:1N156636775;dso:geneOverex:dt:predefinedClass;ec:[1N2].epv:1N1.1N3;et:over:f:186292277;g:928;gt:boxplot;p:1N200015490;pg:1;pvf:8217.8248.16107;scr:datasets:ss:analysis:v:17)

References:

1. Blumenthal A, *et al.* Mechanical stress enhances CD9 expression in cultured podocytes. *American journal of physiology Renal physiology* **308**, F602-613 (2015).
2. Perrot JY, Boucheix C, Mirshahi M, Kazatchkine M, Bariety J. [Monoclonal antibodies against surface antigens of lymphoblasts and blood cells or bone marrow recognize constituents of the human nephron]. *Nephrologie* **5**, 53-57 (1984).
3. Nakamura Y, Handa K, Iwamoto R, Tsukamoto T, Takahasi M, Mekada E. Immunohistochemical distribution of CD9, heparin binding epidermal growth factor-like growth factor, and integrin alpha3beta1 in normal human tissues. *J Histochem Cytochem* **49**, 439-444 (2001).
4. Thorner PS, Ho M, Eremina V, Sado Y, Quaggin S. Podocytes contribute to the formation of glomerular crescents. *Journal of the American Society of Nephrology : JASN* **19**, 495-502 (2008).
5. Smeets B, *et al.* Tracing the origin of glomerular extracapillary lesions from parietal epithelial cells. *Journal of the American Society of Nephrology : JASN* **20**, 2604-2615 (2009).
6. Bollee G, *et al.* Epidermal growth factor receptor promotes glomerular injury and renal failure in rapidly progressive crescentic glomerulonephritis. *Nature medicine* **17**, 1242-1250 (2011).
7. Henique C, *et al.* Nuclear Factor Erythroid 2-Related Factor 2 Drives Podocyte-Specific Expression of Peroxisome Proliferator-Activated Receptor gamma Essential for Resistance to Crescentic GN. *Journal of the American Society of Nephrology : JASN* **27**, 172-188 (2016).

8. Charrin S, Jouannet S, Boucheix C, Rubinstein E. Tetraspanins at a glance. *Journal of cell science* **127**, 3641-3648 (2014).
9. Eymael J, *et al.* CD44 is required for the pathogenesis of experimental crescentic glomerulonephritis and collapsing focal segmental glomerulosclerosis. *Kidney international* **93**, 626-642 (2018).
10. Lenoir O, *et al.* Endothelial cell and podocyte autophagy synergistically protect from diabetes-induced glomerulosclerosis. *Autophagy* **11**, 1130-1145 (2015).
11. Luque Y, *et al.* Endothelial Epas1 Deficiency Is Sufficient To Promote Parietal Epithelial Cell Activation and FSGS in Experimental Hypertension. *Journal of the American Society of Nephrology : JASN* **28**, 3563-3578 (2017).
12. Baraldi A, *et al.* Beta 1 and beta 3 integrin upregulation in rapidly progressive glomerulonephritis. *Nephrology, dialysis, transplantation : official publication of the European Dialysis and Transplant Association - European Renal Association* **10**, 1155-1161 (1995).
13. Prakoura N, Kavvadas P, Kormann R, Dussaule JC, Chadjichristos CE, Chatziantoniou C. NFkappaB-Induced Periostin Activates Integrin-beta3 Signaling to Promote Renal Injury in GN. *Journal of the American Society of Nephrology : JASN* **28**, 1475-1490 (2017).
14. Kagami S, Border WA, Ruoslahti E, Noble NA. Coordinated expression of beta 1 integrins and transforming growth factor-beta-induced matrix proteins in glomerulonephritis. *Laboratory investigation; a journal of technical methods and pathology* **69**, 68-76 (1993).

REVIEWERS' COMMENTS:

Reviewer #1 (Remarks to the Author):

The authors are very responsive. All of my earlier critiques are fully addressed in a satisfactory manner. I have no further concerns and comments.

Reviewer #2 (Remarks to the Author):

I am satisfied with the revised changes

Reviewer #3 (Remarks to the Author):

The authors have nicely addressed my concerns and I don't have any more comments.